# Intracellular sodium elevation reprograms cardiac metabolism

Dunja Aksentijević [1,6], Anja Karlstaedt [2], Marina V. Basalay [1], Brett A. O'Brien [3], David Sanchez-Tatay [1], Seda Eminaga[1], Alpesh Thakker[4], Daniel A. Tennant [4], William Fuller [5], Thomas R. Eykyn [3], Heinrich Taegtmeyer [2] & Michael J. Shattock [1✉]

Intracellular Na elevation in the heart is a hallmark of pathologies where both acute and chronic metabolic remodelling occurs. Here, we assess whether acute (75 μM ouabain 100 nM blebbistatin) or chronic myocardial $Na_i$ load (PLM[3SA] mouse) are causally linked to metabolic remodelling and whether the failing heart shares a common Na-mediated metabolic 'fingerprint'. Control (PLM[WT]), transgenic (PLM[3SA]), ouabain-treated and hypertrophied Langendorff-perfused mouse hearts are studied by [23]Na, [31]P, [13]C NMR followed by [1]H-NMR metabolomic profiling. Elevated $Na_i$ leads to common adaptive metabolic alterations preceding energetic impairment: a switch from fatty acid to carbohydrate metabolism and changes in steady-state metabolite concentrations (glycolytic, anaplerotic, Krebs cycle intermediates). Inhibition of mitochondrial Na/Ca exchanger by CGP37157 ameliorates the metabolic changes. In silico modelling indicates altered metabolic fluxes (Krebs cycle, fatty acid, carbohydrate, amino acid metabolism). Prevention of $Na_i$ overload or inhibition of Na/$Ca_{mito}$ may be a new approach to ameliorate metabolic dysregulation in heart failure.

[1] School of Cardiovascular and Medical Sciences, British Heart Foundation Centre of Research Excellence, King's College London, The Rayne Institute, St Thomas' Hospital, London, UK. [2] Department of Internal Medicine, Division of Cardiology, McGovern Medical School The University of Texas Health Science Center at Houston, Houston TX77030, USA. [3] School of Biomedical Engineering and Imaging Sciences, King's College London, St Thomas' Hospital, London SE1 7EH, UK. [4] Institute of Metabolism and Systems Research, College of Medical and Dental Sciences University of Birmingham, Edgbaston, Birmingham B15 2TT, UK. [5] Institute of Cardiovascular and Medical Sciences, University of Glasgow, Glasgow G12 8TA, UK. [6] Present address: William Harvey Research Institute, Barts and The London School of Medicine and Dentistry, Queen Mary University of London, Charterhouse Square, London EC1M 6BQ, UK. ✉email: michael.shattock@kcl.ac.uk

In the heart, the Na/K ATPase plays a crucial role in transmembrane transport, ion homeostasis, electrical excitability, control of cell volume and contractility[1]. The activity of the Na/K ATPase is regulated by FXYD1, or phospholemman (PLM), the principal sarcolemmal substrate of protein kinases A and C[2,3]. PLM is required for the dynamic control of intracellular sodium ($Na_i$) during increases in heart rate and plays a vital role in $Na_i$ regulation during 'fight or flight'[1]. More recent studies have suggested that cytosolic $Na_i$ regulation also plays an important role in linking mitochondrial Ca-dependent ATP production to mechanical activity and ATP demand[4–6]. Increases in cardiac contractility are largely driven by increases in the cytosolic Ca transient which, in turn, is sensed by mitochondria to ensure that ATP supply matches consumption[7]. In the mitochondria, a rise in matrix calcium ($Ca_{mito}$) activates the $F_1F_0$ ATPase (complex V of mitochondrial respiratory chain)[8], and several Ca sensitive dehydrogenases ($CaDH_{mito}$), including pyruvate dehydrogenase (PDH), α-ketoglutarate dehydrogenase and the NAD-linked isocitrate dehydrogenase (IDH3)[9]. $CaDH_{mito}$ activation of the Krebs cycle results in increased NADH production, which is critical to supply reducing equivalents for the electron transport chain (ETC), free radical (ROS) regulation and redox signalling[5]. This crucial ATP supply-demand relationship has been proposed to be affected by elevated cytosolic $Na_i$, which activates Na/Ca exchange in the inner mitochondrial membrane (NCLX), decreasing $Ca_m$ and leading to impaired $NADH/NAD^+$ redox cycling and metabolic inefficiency[3,4]. In addition to the link with mitochondrial metabolism, the high turnover of ATP required for Na/K ATPase function has been reported to be tightly coupled to ATP supply arising preferentially from glycolysis[10,11] providing further evidence for a link between cytosolic Na ion homeostasis and metabolism.

The rise of $Na_i$ in cardiac pathologies such as heart failure may be due in part to elevated late Na current[12], diastolic Na influx via Na channels[13] and/or reduced NKA function[1]. Cardiac hypertrophy and failure are characterized by elevated $Na_i$[14,15], metabolic insufficiency, increased ROS production and changes in myocardial substrate preference with a shift from dominant fatty acid oxidation towards carbohydrate oxidation[16]. The extent that these changes reflect chronic cellular remodelling, or arise as a consequence of the accompanying $Na_i$ elevation, has not been determined. Whether chronic $Na_i$ elevation, $Na_i$ elevation in hypertrophy or acute $Na_i$ elevation share a common metabolic 'fingerprint', and the details of the metabolic derangements involved also remain largely unknown. The aims of the current study are (i) to determine the detailed metabolic changes associated with chronic Na/K ATPase inhibition in the PLM[3SA] mouse, during cardiac hypertrophy (aortic constriction) and following acute pharmacological $Na_i$ elevation, (ii) to dissociate such changes from the increased inotropic state (and hence metabolic demand) that occurs with $Na_i$ elevation and (iii) to determine whether chronic or acute Na driven changes in metabolism can be reversed by pharmacologically inhibiting mitochondrial NCLX.

We use nuclear magnetic resonance spectroscopy (NMR) to assess $Na_i$ (multiple quantum filtered $^{23}Na$ NMR) and energetics ($^{31}P$ NMR) in real time in isolated perfused mouse hearts. Substrate preference is determined using $^{13}C$ NMR and steady-state metabolomic profile is performed using high-resolution $^1H$ NMR, LC and GC/MS spectrometry. We observe that there are common Na-dependent metabolic alterations in chronic Na/K ATPase inhibition (PLM[3SA] mouse), cardiac hypertrophy and in response to acute $Na_i$ elevation (ouabain). The phenotypes are characterised by a switch in substrate preference from fatty acid to carbohydrate oxidation. However, Krebs cycle fluxes are identified as compensated by in silico modelling while, in the absence of

elevated inotropy, neither chronic nor acute $Na_i$ elevation results in energetic impairment (PCr/ATP) suggesting this to be an adaptive response. The metabolic changes are acutely reversed by inhibiting mitochondrial Na/Ca exchange suggesting a causal link between $Na_i$ elevation and adaptive metabolic remodelling that ensues.

## Results

**Chronic Na elevation alters cardiac metabolic profile in PLM[3SA].** We have used multiple quantum filtered (TQF) $^{23}Na$ NMR spectroscopy to measure intracellular Na in Langendorff-perfused hearts. In the isolated heart, the TQF signal is a composite arising from the sum of various compartments, each with varying electrostatic interactions, Na concentrations, and volumes occupied[17]. In order to define the subcellular origin of this signal we have assessed the relative contribution of subcellular compartments to the total $^{23}Na$ TQF signal (Supplementary Fig. 1). We estimated the following compartmental contributions to the TQF signal: Vascular + extracellular space = 47%. Intracellular signal = 53%, Cytoplasmic signal = 44%, Mitochondrial signal = 9% (Supplementary Note 1).

Using $^{23}Na$ TQF NMR, we have previously shown that $Na_i$ is chronically elevated in hearts from PLM[3SA] mice by ~20–40% compared to WT[17], consistent with an increase in the $K_m$ for $Na_i$ activation of the cardiac Na/K ATPase[17]. Therefore, we first sought to investigate whether basal cardiac metabolism is altered in the PLM[3SA] mouse heart and whether there is a potential link to the elevated $Na_i$ seen in this model. Despite a small alteration in serum glucose and insulin concentration in PLM[3SA] mice (Supplementary Table 2), there was no functional consequence because glucose tolerance tests, glucose transporter expression (GLUT1 and GLUT4) and the skeletal muscle metabolic phenotype (fed and fasted state Supplementary Fig. 2) were comparable between the two genotypes.

Comprehensive metabolic assessment of PLM[3SA] excluded changes in systemic metabolic confounders (circulating metabolites and hormones), liver dysfunction, muscle damage and mitochondrial organization (Supplementary Figs. 2, 3, Supplementary Table 2). There was no difference in the protein expression of IDH3, Ca-sensitive rate determining enzyme of the Kreb's cycle, between the groups (PLM[3SA] $1.0 \pm 0.05$ vs $1.0 \pm 0.07$ AU PLM[WT] $n = 7$/group, Supplementary Fig. 4) and expression of pyruvate dehydrogenase (PLM[3SA] $1.03 \pm 0.1$ vs $1 \pm 0.08$ PLM[WT] $n = 7$/group, Supplementary Fig. 4).

Furthermore, there were no differences in ex vivo function between WT and PLM[3SA] mouse hearts paced at 550 beats min[−1] (LVDP, HR, coronary flow) or non-paced hearts (Supplementary Fig. 5) in agreement with previous in vivo observations in the same model[18]. Previous studies indicate an absence of inotropic changes in PLM[3SA] mouse hearts, despite the elevation of intracellular Na[18], which is potentially driven by adaptations in sarcolemmal Na regulatory protein expression (NCX and PLM/Na/K ATPase α2 subunit).

Given the lack of overt systemic metabolic phenotype in PLM[3SA] we next performed $^{13}C$ NMR analysis of substrate contribution to oxidative metabolism, revealing a switch in substrate preference from fatty acid to glucose oxidation (Fig. 1a). $^1H$ NMR metabolite analysis of extracted heart tissue revealed a significant global depletion of energy providing substrates including lactate, amino acids (aspartate, glutamine, glutamate, alanine, glycine), intermediates of Krebs cycle (citrate, succinate) and constituents of lipid metabolism (phosphocholine, acetyl carnitine, acetate) (Fig. 1c). PLM[3SA] hearts did not display a compromised PCr/ATP ratio which was the same as their WT counterpart (Fig. 1e)

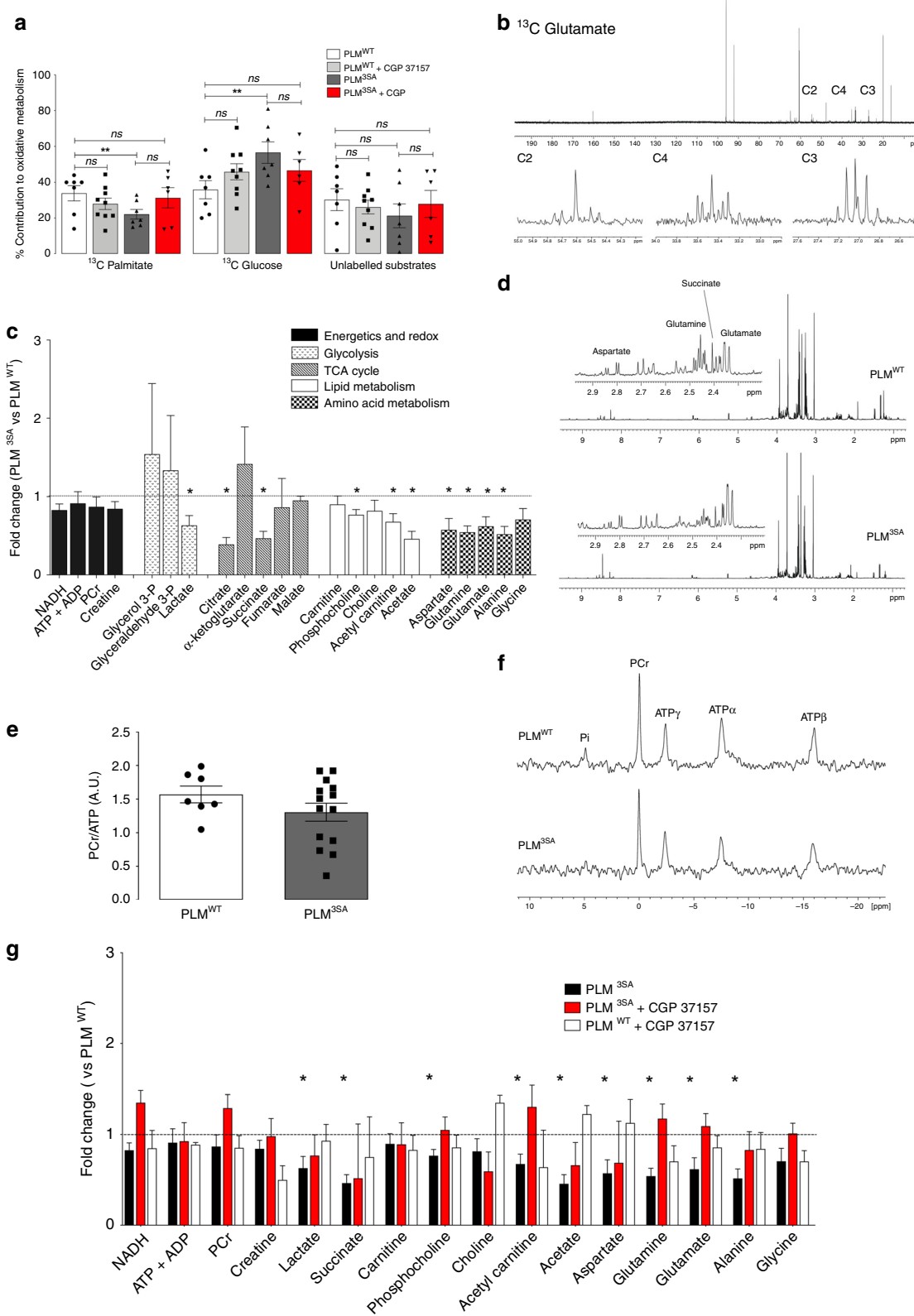

indicating that the switch from fatty acid to glucose oxidation and depletion of metabolic intermediates is adaptive. To explore whether the metabolic alterations in this model were causally linked to the elevation in Na$_i$ we next explored the influence of inhibiting NCLX. Treatment with 1 μM CGP37157 was without effect in wild-type mice but had a significant effect on the altered cardiac metabolic profile in transgenic PLM$^{3SA}$ hearts. It normalized concentrations of depleted metabolic intermediates (Fig. 1g) and normalized rates of PLM$^{3SA}$ palmitate oxidation vs. WT controls (Fig. 1a), thus reverting the glucose-palmitate substrate switch in 30 min of exposure. In the absence of systemic metabolic derangement (whole body-muscle, liver, serum etc), hypertrophy, cardiac dysfunction or increased inotropy, our data suggest that chronic Na$_i$ elevation

**Fig. 1 Metabolic profile of PLM$^{3SA}$ and PLM$^{WT}$ hearts. a** Myocardial $^{13}$C NMR assessment of oxidative metabolism—% contribution of $^{13}$C-U palmitate, $^{13}$C 1,6 glucose and the remnant unlabelled $^{13}$C substrate pool (triglycerides, glycogen, pyruvate, lactate, ketone bodies). $^{13}$C NMR assessment of oxidative metabolism after 30 min administration of CGP37157 in $^{13}$C-KH$_{metab}$ buffer—PLM$^{3SA}$ ($n = 6$) vs PLM$^{WT}$ ($n = 7$) hearts. (*$P < 0.03$ PLM$^{3SA}$ palmitate oxidation vs PLM$^{WT}$, $P < 0.02$ PLM$^{3SA}$ glucose oxidation vs PLM$^{WT}$, $t = 3.7$ df = 8). **b** Representative $^{13}$C spectrum from perfused mouse heart: full $^{13}$C glutamate spectrum. Multiplet peak patterns of $^{13}$C glutamate C2, C4, C3 glutamate resonances. **c** Metabolomic profile—metabolite fold change normalized to control concentration (PLM$^{WT}$ levels = 1) with propagated errors (SEM). $^{1}$H NMR metabolomic analysis: NAD, ATP + ADP, PCr, creatine, carnitine, phosphocholine, choline, acetyl carnitine, acetate, aspartate, glutamine, glutamate, glycine, alanine. GS-MS/MS analysis: pyruvate, lactate, citrate, isocitrate, α-ketoglutarate, succinate, fumarate, malate (PLM$^{3SA}$ $n = 5$, PLM$^{WT}$ $n = 8$; lactate PLM$^{3SA}$ $n = 12$, PLM$^{WT}$ $n = 10$, succinate, glutamate PLM$^{3SA}$ $n = 14$ PLM$^{WT}$ $n = 10$, aspartate PLM$^{3SA}$ $n = 13$ PLM$^{WT}$ $n = 10$) (*$P < 0.05$ vs PLM$^{WT}$, succinate, citrate, glutamine, alanine $P < 0.005$ vs PLM$^{WT}$ by $t$-test, two tailed, df = 10). **d** Representative $^{1}$H metabolomic NMR spectra. **e** Myocardial energetic reserve (PCr/ATP) ratio determined by $^{31}$P NMR spectroscopy (PLM$^{3SA}$ $n = 14$, PLM$^{WT}$ $n = 7$, $P < 0.22$, vs control by $t$-test, two tailed, $t = 1.2$, df = 19). **f** Representative $^{31}$P spectra of PLM$^{WT}$ and PLM$^{3SA}$ hearts. **g** Impact of 30 min perfusion with 1μM CGP37157 in KH$_{metab}$ buffer on $^{1}$H NMR metabolomic profile of PLM$^{3SA}$ versus PLM$^{WT}$ hearts. Metabolite fold change vs control (control = 1) with propagated error of mean. ($n = 8$/group PLM$^{3SA}$ and PLM$^{WT}$). **$P < 0.05$ vs PLM$^{WT}$, succinate, citrate, glutamine, alanine $P < 0.005$ vs PLM$^{WT}$ by $t$-test, two tailed, df = 10. Data are mean ± SEM. Source data are provided as a Source Data file.

(PLM$^{3SA}$ vs PLM$^{WT}$) leads to significant changes in myocardial metabolism.

**Chronic Na elevation alters metabolism in cardiac hypertrophy.** We next sought to establish whether the chronic alterations in metabolism seen in the PLM$^{3SA}$ mouse are recapitulated in pathological hypertrophy. Myocardial hypertrophy was induced in C57BL/6J mice by pressure overload following suprarenal aortic constriction (banding) for 5 weeks (Fig. 2a, Supplementary Table 3). Cardiac hypertrophy was associated with a significant decline in in vivo function (Supplementary Table S3) and, as previously demonstrated, a 40% increase in myocardial Na$_i$[15]. There was no evidence of heart failure or cardiac decompensation as indicated by the absence of pulmonary oedema (Supplementary Table 3). Ex vivo function of unpaced hypertrophied hearts was not significantly different to sham controls. Consistent with previous studies, we found that LVDP was impaired when hearts were paced at higher heart rates (550 beats m$^{-1}$)[18]. $^{13}$C NMR analysis of substrate contribution to oxidative metabolism revealed that pressure overload hypertrophy was also associated with a switch in substrate preference from fatty acid to glucose oxidation (Fig. 2b) as previously shown[19], while the contribution of pyruvate, lactate and endogenous substrates was decreased. As to the PLM$^{3SA}$ data, tissue NAD, PCr, aspartate and alanine concentrations were increased while all other measured metabolite concentrations were unchanged compared to sham (Krebs cycle, intermediates, lipid and amino acid metabolism constituents) (Fig. 2c). Further, the intracellular PCr/ATP ratio was preserved (PCr/ATP Banded 1.17 ± 0.07 vs Sham 1.21 ± 0.18 $n = 5$/group) suggesting that metabolic remodelling in the heart remains adaptive after five weeks of pressure-overload induced hypertrophy. We next explored whether the metabolic alterations observed in hypertrophy might also be linked to the elevation in Na$_i$ via the action of NCLX. Treatment with 1 μM CGP37157 also had a significant effect on substrate preference in hypertrophic hearts, leading to an increase in palmitate oxidation and a decrease in glucose oxidation, thus reverting the glucose-palmitate substrate switch (Fig. 2b). 1-μM CGP37157 treatment had a less pronounced effect on the steady-state metabolite profile in the hypertrophy group due to the more subtle changes observed in this group (Fig. 2d).

**Acute Na$_i$ elevation alters cardiac metabolic profile without compromising energetics.** Acute pharmacological inhibition of Na/K ATPase in wild-type (WT) mouse hearts using ouabain (Ouab) caused a 140% increase in intracellular Na (Fig. 3a) and an associated 70% increase in LVDP within 30 min (Fig. 3b). Myocyte

protein expression (IDH3, pyruvate dehydrogenase) was not altered following 30 mins of drug treatment (Supplementary Fig. 4). Central to our hypothesis, and to be consistent with the PLM$^{3SA}$ and hypertrophy groups, the direct effects of Na$_i$ on cardiac metabolism need to be assessed independently of changes in inotropy and energy demand. Accordingly, in pilot experiments, we defined a concentration of 100 nmol l$^{-1}$ for the myofilament uncoupler blebbistatin (Blebbi) that, when simultaneously perfused with ouabain, was sufficient to antagonise the positive inotropy without itself causing a negative inotropy. The inotropically neutral combination of 75 μmol l$^{-1}$ ouabain and 100 nmol l$^{-1}$ blebbistatin was then used throughout in ex vivo mouse heart experiments. None of the pharmacological combinations affected heart rate (Fig. 3c) or coronary flow (Fig. 3d) over the 50 min course of perfusion. $^{13}$C NMR analysis of substrate contribution to oxidative metabolism also revealed that acute ouabain/blebbistatin treatment resulted in reduced oxidation of fatty acids in relation to the contribution of other fuels, principally carbohydrates (glucose, pyruvate, lactate) (Fig. 4a). Acute Na elevation also resulted in increased levels of lactate and isocitrate, as well as the big drop-off in metabolite levels of Krebs cycle intermediates downstream from α-ketoglutarate dehydrogenase (succinate, fumarate, malate, Fig. 4b). These effects are consistent with reduced Ca-dependent activation of the critical Krebs cycle dehydrogenases at these points.

Using $^{1}$H NMR and GC- and LC-MS/MS metabolomic profiling, we identified extensive metabolic changes in intermediary metabolites including the Krebs cycle, oxidative phosphorylation, glycolysis and anaplerosis (Fig. 4b) that was more pronounced than that observed in the PLM$^{3SA}$ group. Acute Na$_i$ elevation, in the absence of any change in inotropy did not compromise myocardial energetics (PCr/ATP ratio, Fig. 4c). Lowering contractility below baseline with Blebbi alone resulted in a modest PCr/ATP elevation (Fig. 4c). Treatment with 1-μM CGP37157 had a significant effect on cardiac metabolic profile during acute Na$_i$ elevation, normalizing the concentrations of metabolic intermediates previously seen to be depleted (Fig. 4d). In the absence of increased inotropy, our data suggest that acute Na$_i$ elevation leads to significant changes in myocardial metabolism.

**In silico modelling reveals common Na-induced metabolic changes.** To assess the impact of Na$_i$ on cardiac metabolism at a systems scale, we conducted mathematical modelling using CardioNet[20,21]. CardioNet has been successfully applied to identify limiting metabolic processes and estimate flux distributions[22]. We determined flux distributions using flux balance analysis (FBA), which seeks to determine flux rates during steady-state while optimising an objective function (see Supplementary Methods for

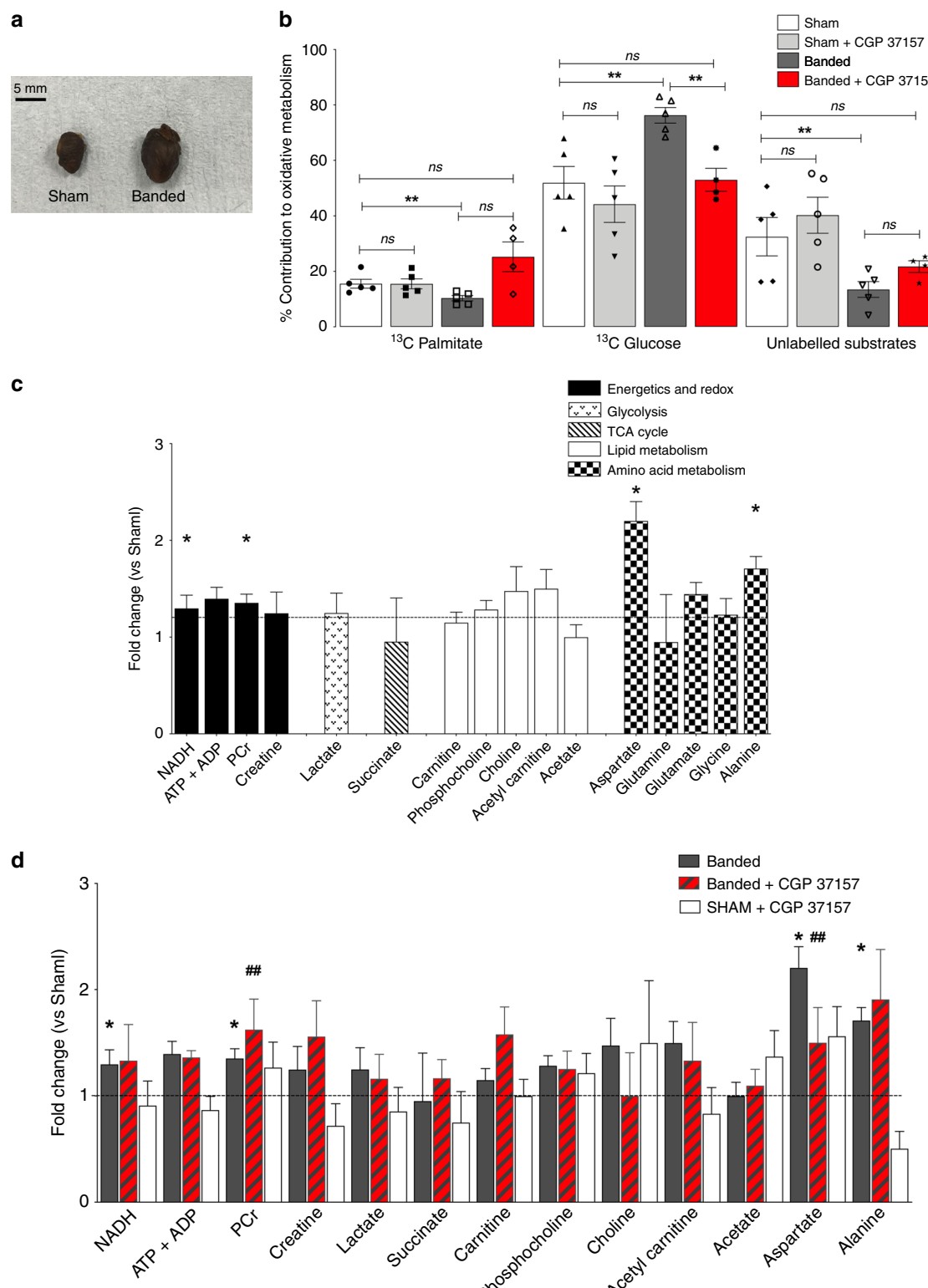

details). Our experimental data show that while Na elevation re-programs metabolism, it does so while not compromising energetics—as demonstrated by our $^{31}$P-NMR measurements showing maintained ATP, PCr, PCr/ATP ratios and pH$_i$. Therefore, we defined the optimisation problem to maximize ATP provision within a set of constraints defined by our experimental conditions. We included experimentally measured metabolic data during acute and chronic Na$_i$ elevation into the simulations to

determine which metabolic pathways are consistent with the cardiac adaptation during our experimental conditions. These simulations were conducted without constraining enzymatic activities; thus the optimisation problem was defined by experimentally determined metabolite levels. Our modelling did not include Ca-dependence as a variable, because electrolytes are not part of the Cardionet metabolic network. Principal component analysis (PCA) (Supplementary Fig. 7) of estimated flux

**Fig. 2 Metabolic profile of pressure-overload induced (banding) hypertrophy. a** representative Sham control and banded hypertrophy (dry hearts).
**b** Myocardial $^{13}$C MRS assessment of oxidative metabolism—% contribution of $^{13}$C-U palmitate, $^{13}$C 1,6 glucose and the remnant unlabelled $^{12}$C substrate pool (triglycerides, glycogen, pyruvate, lactate, ketone bodies). Impact of the mitochondrial Na/Ca exchange inhibition with CGP37157 on metabolic fluxes: myocardial $^{13}$C NMR assessment of oxidative metabolism after 30 min administration of CGP37157 in $^{13}$C-KH$_{metab}$ buffer—% contribution of $^{13}$C-U palmitate, $^{13}$C 1,6 glucose and the remnant unlabelled $^{12}$C substrate pool to oxidative phosphorylation Banded vs Sham hearts ($n = 5$/group). Banded vs Sham palmitate oxidation $P < 0.05$, glucose oxidation $P < 0.005$, unlabelled $P < 0.03$ by $t$-test (two-tailed); Banded + CGP37157 vs banded $P < 0.05$, by one-way ANOVA and Bonferroni multiple comparisons post-test. **c** $^{1}$H NMR metabolite profile - fold change normalized to control concentration (Sham levels = 1) with propagated error (SEM) ($n = 6$/group). *$P < 0.05$ vs control by $t$-test (two tailed, df = 9). **d** Impact of 30 min perfusion with 1 μM CGP37157 in KH$_{metab}$ buffer on $^{1}$H NMR metabolomic profile of Banded ($n = 4$) versus Sham ($n = 5$) hearts. *$P < 0.05$ vs Sham, by t-test (two tailed) ##$P <$ 0.01 PCr vs. Sham + CGP37157, $P < 0.01$ Aspartate Banded CGP37157 vs Sham by one-way ANOVA and Bonferroni multiple comparisons post-test. Data are mean ± SEM. Source data are provided as a Source Data file.

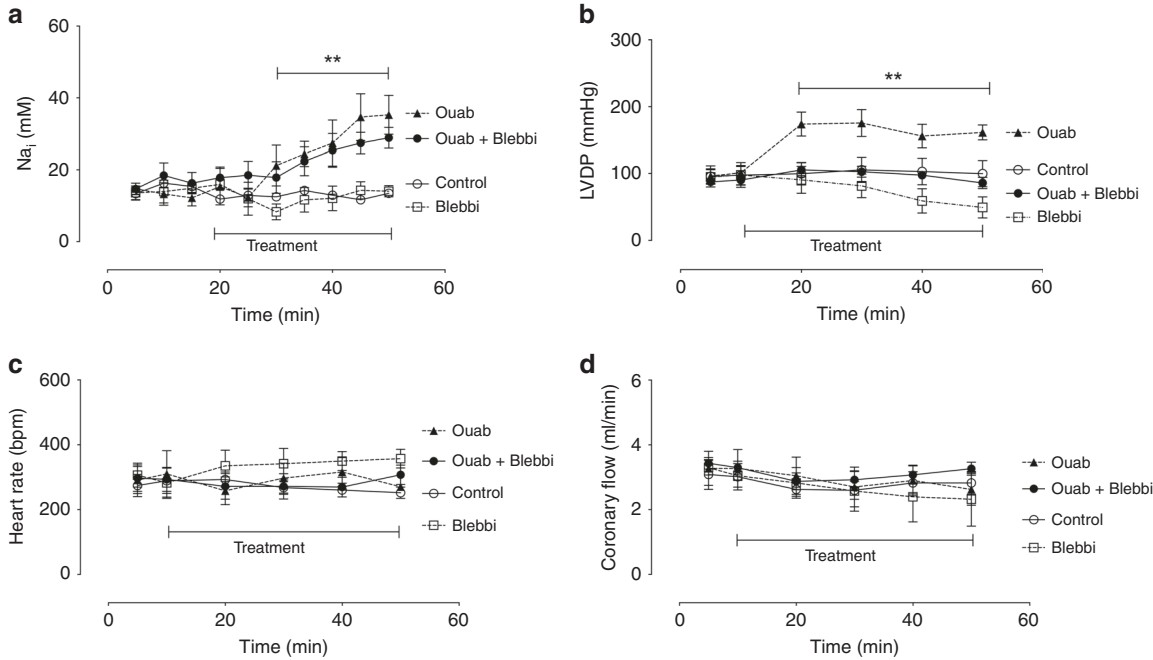

**Fig. 3 Impact of acute 30 min Na/K ATPase inhibition on Na$_i$ and ex vivo function. a** Time course of Na$_i$ elevation measured by $^{23}$Na TQF filtered NMR. 40 min $P < 0.01$ vs baseline, 45 and 50 min $P < 0.0001$ vs baseline. **b** Impact on left ventricular developed pressure (LVDP) **$P < 0.01$ versus baseline. **c** Heart rate (HR) and **d** coronary flow. Data are presented as mean ± SEM. Ouab = 75 μmol/l ouabain ($n = 8$); Ouab + Blebbi = 75 μmol/l ouabain + 100 nmol/l blebbistatin ($n = 11$); Control = KH buffer + vehicle (DMSO) ($n = 7$); Blebbi = 100 nmol/l blebbistatin ($n = 5$). $P$ value is for the effect of the drug treatment by two-way ANOVA. Source data are provided as a Source Data file.

distributions clearly classified samples according to their Na$_i$ elevation status. Unsupervised hierarchical cluster analysis (Fig. 5) together with annotation enrichment of the observed clusters (Supplementary Fig. 8) showed that Na$_i$ elevation affected a multitude of metabolic pathways, and therefore defined a possible metabolic hallmark of Na$_i$ elevation irrespective of its duration or aetiology. Key reactions involved in glycolysis, Krebs cycle and OXPHOS reactions were upregulated ($P < 0.05$; Fig. 5). Likewise modelling estimated an enhanced amino acid, phospholipid and purine metabolism while predicting a reduction in the metabolism of fatty acids, ketone bodies and nucleotides (Fig. 6). Interestingly, in acute Na$_i$ overload, the model identified that a reduction in fatty acid utilisation created extensive need for enhanced metabolism of pyruvate, lactate and amino acids to maintain normal Krebs cycle and oxidative phosphorylation flux. In chronic Na$_i$ overload, the model identified that reduction of fatty acid oxidation enhanced glucose utilisation and ketone metabolism leading to increased fluxes through the Krebs cycle (including CaDH$_{mito}$) and oxidative phosphorylation (Fig. 6). Together our in silico modelling showed that elevation of Na$_i$ promotes a shift from fatty acid towards glucose oxidation and extensive metabolic flux remodelling resulting in maintained ATP

synthesis. We validated our model by comparing the experimentally determined metabolic profile to our simulations. Mathematically predicted metabolite changes correlate with experimentally determined changes ($R^2 = 0.72$; Pearson correlation coefficient, $r = 0.85$; Supplementary Fig. 9). We conclude that our mathematical simulations recapitulated the experimentally observed metabolic profile—in particular regarding the shift from fatty acid ß-oxidation towards carbohydrate oxidation—while expanding our understanding of cardiac metabolic adaptation in response to Na$_i$ elevation.

## Discussion
We hypothesised that elevated myocardial Na$_i$ may be a unifying mechanism leading to myocardial metabolic remodelling regardless of its aetiology (pharmacological, transgenic or pathological hypertrophy–mediated Na/K ATPase inhibition) or duration (acute-30 min, chronic-5 or 25 week Na$_i$ overload) that precedes the onset of energetic deficit and functional deterioration. We used an experimental approach combining retrogradely perfused mouse hearts under physiological conditions (temperature, $O_2$ availability, pH, unrestricted metabolic substrate availability) with continuous non-invasive NMR ($^{23}$Na and $^{31}$P),

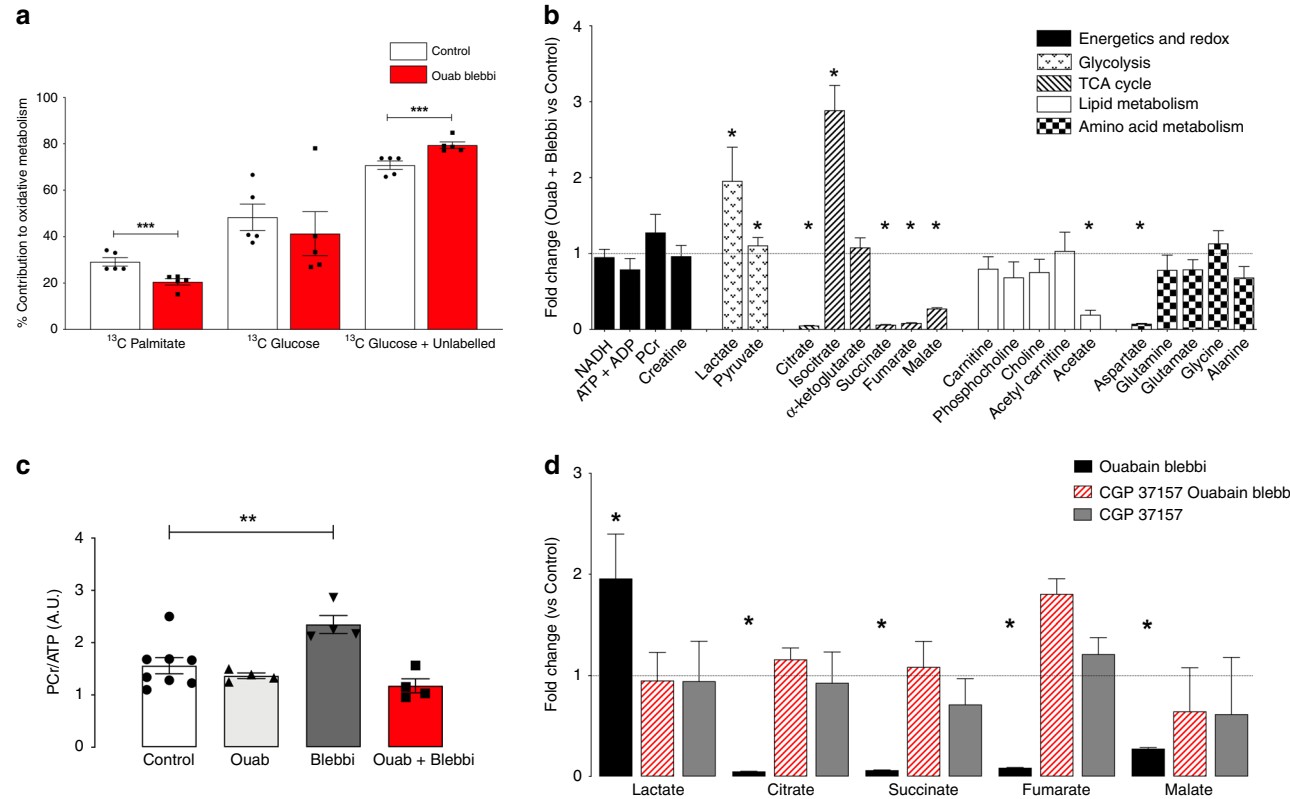

**Fig. 4 Impact of acute 30 min $Na_i$ elevation on myocardial metabolism. a** Myocardial $^{13}C$ NMR assessment of oxidative metabolism—% contribution of $^{13}C$-U palmitate, $^{13}C$ 1,6 glucose and the remnant unlabelled $^{13}C$ substrate pool (pyruvate, lactate, amino acids, triglycerides, glycogen). $^{13}C$ glucose + unlabelled = 100%-$^{13}C$ palmitate oxidation) ($n = 5$/group) ***$P < 0.005$ vs control by unpaired t-test (two-tailed, $t = 3.74$ df = 8). **b** Metabolic profile plotted as metabolite fold change normalized to control concentration (control levels = 1) with propagated error (SEM). $^1H$ NMR metabolite analysis: NAD, ATP + ADP, PCr, creatine, carnitine, phosphocholine, choline, acetyl carnitine, acetate, aspartate, glutamine, glutamate, glycine, alanine. GC and LC-MS/MS analysis: pyruvate, lactate, citrate, isocitrate, α ketoglutarate, succinate, fumarate, malate. C57/BL6 hearts perfused for 30 min with Control = Krebs Henseleit buffer + vehicle (DMSO); Ouab = 75 μmol/l ouabain; Blebbi = 100 nmol/l blebbistatin; Ouab + Blebbi = 75 μmol/l ouabain + 100 nmol/l ouabain. $P < 0.005$ vs control by unpaired t-test (two-tailed, df = 2.2). **c** Energetic reserve PCr/ATP (Control $n = 8$, $n = 4$/treatment group, *$P < 0.001$ vs control by one-way ANOVA $F = 8.6$). **d** Impact of the mitochondrial Na/Ca exchange inhibition with CGP37157 on GC-MS/MS metabolic profile. CGP37157 + Ouab + Blebbi = 1 μmol/l CGP37157 + 75 μmol/l ouabain + 100 nmol/l blebbistatin; CGP37157 = 1 μmol/l CGP37157; Control = KH buffer + vehicle (DMSO); Blebbi = 100 nmol/l blebbistatin ($n = 5$/group)*$P < 0.05$ lactate, $P < 0.0002$ citrate, succinate, fumarate, malate vs control by one-way ANOVA. Data are mean ± SEM (**a**, **c**) and fold change versus control (control = 1) with propagated error of mean (**b**, **d**). Comparisons by one-way ANOVA were subject to Bonferroni multiple comparisons post-test. Source data are provided as a Source Data file.

steady-state $^{13}C$ NMR analysis of substrate utilization and flux, high-resolution $^1H$ NMR metabolic profiling, traditional in vitro biochemical methods (LC GC MS/MS) as well as in silico computer modelling to assess the impact of regulating $Na_i$ (via Na/K ATPase inhibition) on mitochondrial ATP provision.

Chronic cardiac $Na_i$ elevation in the PLM$^{3SA}$ mouse was associated with adaptations in cardiac metabolism with a switch in substrate preference from oxidation of fatty acids to carbohydrates (glucose)[23]. This switch appears to be adaptive in the chronic situation as high-energy phosphate and redox metabolite pools (PCr/ATP, NADH) are maintained. However, our results suggest that in the long term, metabolic remodelling in response to chronically elevated $Na_i$ may be insufficient to compensate, indicated by a sustained depletion of metabolic substrates such as lactate and amino acids[24,25] and reduced intracellular lipid metabolism compared to WT mice. Na/Ca$_{mito}$ inhibition by CGP37157 normalized the substrate utilisation profile (palmitate oxidation) as well as normalising the levels of depleted metabolites in PLM$^{3SA}$ hearts with chronically elevated $Na_i$.

Our study shows that Na-elevation does not compromise energy supply as evidenced by conserved concentration of ATP and PCr, as well as, ATP/PCr ratios. In concordance with our

experimental studies, mathematical modelling using CardioNet predicted remodelling of cardiac metabolism during chronic $Na_i$ elevation, maintenance of normal ATP provision through increased glucose oxidation and anaplerotic replenishment of Krebs cycle flux. It shows that the impact of impaired $Na_i$ homeostasis on mitochondrial ATP production is mechanistically more complex than previous studies suggested using isolated cells and organelles and that there is no evidence of an energetic deficit.

When cytoplasmic Na was chronically elevated, Na/Ca$_{mito}$ inhibition by CGP37157 normalised the altered substrate utilization profile (palmitate oxidation) and normalising the levels of depleted anaplerotic metabolites in PLM$^{3SA}$ hearts. However, when CGP37157 was administered to control hearts (in the absence of chronic Na elevation) it had only a negligible effect on cardiac metabolism (Figs. 1 and 2) and function (Online Supplement Fig. 6). While there is a wealth of literature (including in the heart) showing that CGP37157 inhibits NCLX when the system is perturbed, when matrix Ca is measured using mitochondrially targeted aequorin, CGP37157 does not increase basal matrix Ca[26]. This suggests that lowering NCLX activity below basal conditions does not do the opposite to increasing the

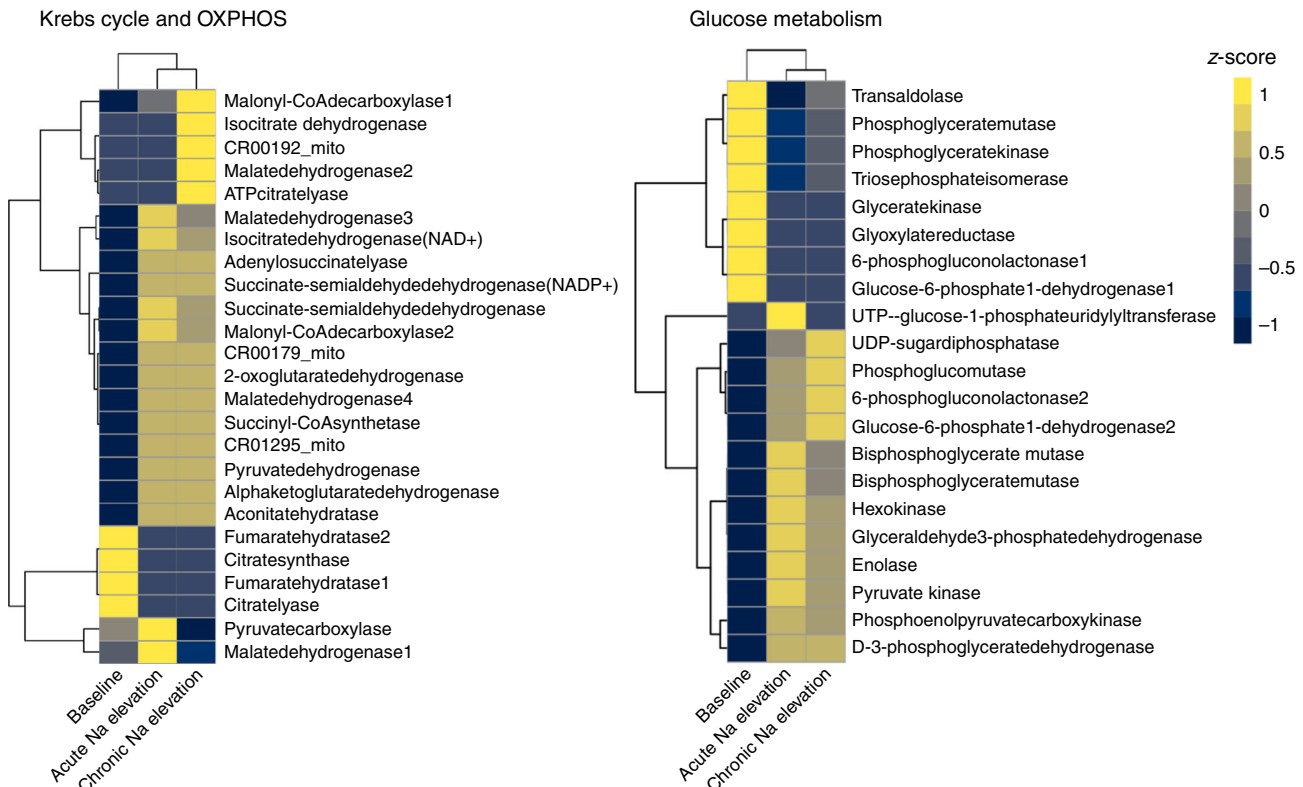

**Fig. 5 Metabolic adaptation in response to acute and chronic Na elevation.** Unsupervised hierarchical clustering of estimated z-scored flux rate changes reveals metabolic adaptation in response to [Na]$_i$ elevation. Heat maps summarize results for reactions in the Krebs cycle, OXPHOS, and glucose metabolism. Flux distributions were calculated by Flux balance analysis (FBA) using the mammalian network of cardiac metabolism, CardioNet. Z-scores were calculated to visualize how many standard deviations an estimated flux rate is away from the mean across all experimental groups. The z-score describes the distance from the mean for a given flux rate as a function of the standard deviation. For example, a z-score equal to 1 represents a flux for a given experimental group that is 1 standard deviation greater than the mean across all experimental groups. The colour scale indicates the degree to which estimated flux rate changes are predicted to be respectively lower or higher in response to Na$_i$ elevation. Source data are provided in Supplementary Data 1.

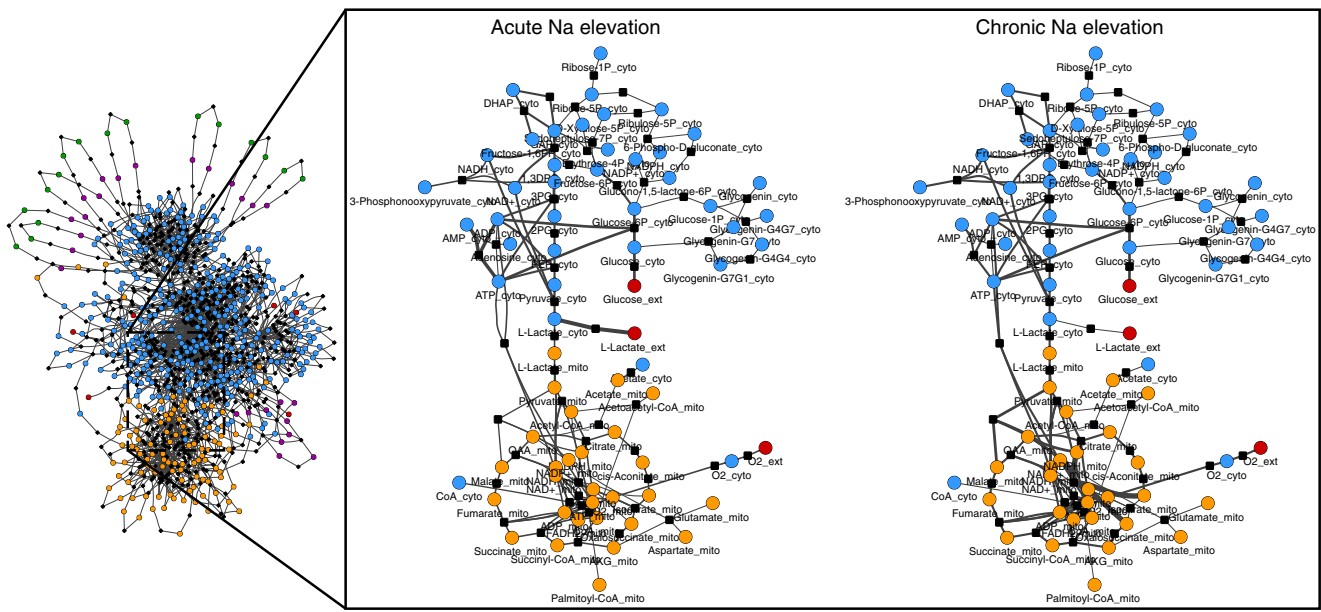

**Fig. 6 CardioNet in silico metabolic flux changes in response to acute and chronic Na$_i$ elevation.** Graph denotes estimated flux distribution in response to acute and chronic Na$_i$ elevation. The coloured nodes represent metabolites assigned to five different compartments: extracellular space, cytosol, mitochondria, microsome, lysosome. The black square nodes indicate reactions; two reactions are linked by a directed edge indicating the reaction flux. The line thickness of each edge is proportional to predicted flux rate change. Source data are provided as a Source Data file.

activity under Na-loaded conditions perhaps due to a non-linear relationship between NCLX and intra-mitochondrial Ca.

Pressure-overload induced hypertrophy by aortic banding resulted in significant hypertrophy, in vivo dysfunction and myocardial $Na_i$ overload after five weeks. [13]C assessment of oxidative substrate utilization in hypertrophied perfused hearts revealed a substrate switch from palmitate to glucose oxidation in keeping with the literature and of similar magnitude to that observed in the chronic Na elevation in PLM[3SA] hearts[16,27]. Similarly to PLM[3SA], there was no evidence of energetic impairment or increase in anaerobic glycolysis. However, a general decline in energy substrate metabolism is not a uniform finding in all hypertrophy and heart failure models[19] including ours, highlighting involvement of alternative metabolic aetiologies. Our study revealed that metabolic remodelling occurs during chronic hypertrophy induced $Na_i$ overload independent of structural and functional remodelling, and this remodelling causes a substrate switch favouring utilization of glucose over fatty acids. $Na/Ca_{mito}$ inhibition by CGP37157 normalized the substrate utilization profile (reversal to palmitate as the main oxidative fuel) in hypertrophic hearts. Collectively, these observations suggest that the early prevention of myocardial $Na_i$ elevation (and consequent decline in mitochondrial matrix Ca) has the potential to reverse metabolic derangement in pathological hypertrophy by preventing deterioration in energy substrate utilization, oxidative phosphorylation and high-energy phosphate transfer.

Acute $Na_i$ elevation via pharmacologic Na/K ATPase inhibition with ouabain, in the presence of blebbistatin, resulted in increased $Na_i$ combined with the most severe metabolic remodelling observed in our study: depletion of Krebs cycle intermediates including citrate, succinate, fumarate, malate and a substrate switch from fatty acid oxidation to increased use of other fuels, primarily carbohydrates (pyruvate, lactate). There was no compromise in levels of high-energy phosphates nor in the redox state. No changes in coronary flow, PCr/ATP ratio nor succinate tissue levels were observed, indicating that acute $Na_i$ elevation does not cause ischaemia nor energetic compromise. Experimental findings were consistent with computational estimation of the metabolic changes using CardioNet[21]. With the contribution of fatty acid β-oxidation to ATP provision decreased, simulations showed that ATP provision for contractile work is maintained due to compensatory increase of amino acid and carbohydrate utilisation (pyruvate, lactate) to replenish Krebs cycle intermediates. Maintained contractile function and ATP provision in the face of high $Na_i$ comes at the cost of increased metabolic substrate utilization and enhanced fluxes leading to significant metabolite depletion. These findings are in keeping with previously reported features of metabolic inefficiency arising from simultaneously increased yet mismatched metabolic fluxes and a failure of fatty acid β-oxidation to compensate[28,29]. Administration of the $Na/Ca_{mito}$ exchanger antagonist CGP37157 in acute $Na_i$ overload completely abrogated the metabolic changes (intracellular metabolite levels and Krebs cycle intermediate concentrations) highlighting the importance of cytosolic $Na_i$ in regulating mitochondrial matrix Ca. However, our data show that in the perfused beating heart acute $Na_i$ elevation resulted neither in decreased oxidative metabolic flux nor ATP and NADH depletion, thus the impact of impaired $Na_i$ homeostasis on mitochondrial ATP production is mechanistically more complex than previous studies suggested using isolated cells and organelles[30,31].

Historically, CGP37157 with its submicromolar inhibitory potency ($IC_{50}$ 0.4 μM) has been suggested as a candidate for the augmentation of ATP reserves in heart failure, insulin-dependent diabetes mellitus and acute myocardial infarction via $Na/Ca_{mito}$

inhibitory mechanism[32,33]. In vivo studies utilizing chronic CGP37157 delivery in a guinea pig model of sudden cardiac death and heart failure suggested anti-arrhythmic potential of the drug[34]. However, in separate cohort studies concerns were raised regarding the inhibitory effect on insulin secretion[35,36] as non-specific effects on microsomal Ca flux (ryanodine receptor/ SERCA) and other Ca channels and transporters in myocytes[35,37]. Therefore, further characterization of CGP37157 or more selective $Na/Ca_{mito}$ inhibitors would be required before considering this approach to the treatment of $Na_i$ induced-metabolic remodelling in cardiac pathologies.

How much of CGP37157 actually reaches the mitochondria and how that translates into $Na/Ca_{mito}$ inhibition is unknown due to a lack of direct way to measure $Na/Ca_{mito}$ inhibition in an intact heart. Given the lethality of the $Na/Ca_{mito}$ KO[38], complete inhibition of NCLX may be catastrophically detrimental. However, in our studies it seems that partial, transient inhibition is survivable in the short term. It is clear, however, that even if the NCLX inhibition is partial, it is sufficient to acutely prevent the metabolic switch. Given CGP37157 would treat a secondary effect of elevated $Na_i$ rather than a primary cause, our study suggests that in order to ameliorate metabolic dysfunction in pathological hypertrophy, pharmacological activation of Na/K ATPase to address $Na_i$ overload may be a superior therapeutic strategy with fewer side effects[2,12]. The outcome of our study also highlights the likely unsuitability of cardiac glycosides (digoxin) for the treatment of hypertrophy and heart failure as they elevate $Na_i$ and would thus cause rather than prevent metabolic remodelling.

This study shows that cytoplasmic Na elevation, irrespective of its aetiology or duration, is instrumental in driving metabolic changes, including substantive reprogramming of substrate metabolism, without compromising energetics. These changes are blocked by an inhibitor of mitochondrial Na/Ca exchange suggesting that this reprogramming involves Na-dependent changes in mitochondrial matrix Ca as the end effector. In the present study, mitochondrial matrix Ca was not measured nor have key Ca-sensitive enzymes been identified, thus the exact nature of such interactions remain to be determined.

A feature of this reprogramming is that it does not appear to compromise energy supply as ATP, PCr and ATP/PCr ratios are conserved. Our study-design investigating acute ouabain exposure attempts to keep ATP demand constant by using blebbistatin titration to limit the induced inotropy. However, while in the beating myocardium, myosin ATPase is responsible for the vast majority of ATP consumption (76%), SERCA and Na/K ATPase also contribute (15% and 9% respectively)[39]. In the present experiments, energy consumption through these pathways may therefore slightly increase but this does not appear to reach the level where ATP supply and energetics are compromised.

Lastly, mathematical modelling of complex biological systems is biased towards known enzymes and metabolites. CardioNet does not include simulations of electrolyte sensitivities and consequently CardioNet cannot identify specific Ca or Na-sensitive enzymes or model their response to ionic changes. Nevertheless, the model does allow pathways to be identified at a systems-wide level that are altered in response to acute and chronic Na elevation based on measured experimental metabolic profiles.

Myocardial $Na_i$ overload leads to a complex series of metabolic perturbations including a switch in substrate reliance from fatty acid oxidation to increased reliance on carbohydrates for ATP provision, with a predicted alteration in mitochondrial metabolic fluxes at the expense of metabolic coupling and a resulting depletion of metabolite pools. These changes appear to be independent of the cause of $Na_i$ elevation, duration of the Na/K ATPase inhibition or pathological origin. The metabolic alterations precede energetic and functional impairment in the heart

thus preventing a $Na_i$ induced mismatch between ATP supply and demand. Alterations in $Na/Ca_{mito}$ exchanger activity by cytosolic $Na_i$ overload contribute to the observed metabolic changes as treatment with the antagonist CGP37157 ameliorates the metabolic adaptations observed. The dynamics of metabolic remodelling due to altered $Na_i$ occur before structural and functional remodelling of the failing heart. Whether these adaptations become maladaptive under stress warrants further investigation.

## Methods

**Animals**. Hearts were isolated from male C57BL/6J male mice (~20–25 g) (Charles River JAX[TM] stock number 000664) or from WT and PLM[3SA] knock-in mice[18]. This investigation complied with all the relevant ethical regulations for animal testing and research: UK Home Office Guidance on the Operation of the Animals (Scientific Procedures) Act, 1986.

PLM[3SA] knock-in mice were backcrossed with C57BL/6J mice (Charles River, UK) for >5 generations and were generated by heterozygous pair mating. The PLM[3SA] mouse expresses a non-phosphorylatable form of PLM in which Ser 63, 68, and 69 have been mutated to alanine and exhibits a Na/K ATPase that is unresponsive to kinase regulation and hence shows chronically elevated $Na_i$[15]. Unless otherwise stated, littermates were used as the appropriate wild-type controls (PLM[WT]). Animals were kept under pathogen-free conditions, 12-h light–dark cycle, controlled humidity (~40%), temperature (20–22 °C), and fed chow and water ad libitum. All animals used in studies were male. For pharmacologically induced acute $Na_i$ elevation studies, 6-week-old C57BL/6J male mice (~25 g body weight) were purchased from Charles River (UK). Myocardial hypertrophy was induced in 6-week-old C57BL/6J mice (20–22 g) (Charles River, UK). For cellular compartmentation of the $^{23}$Na TQF NMR signal experiment, Male Wistar rats (250 g) were purchased from Charles River, UK.

**Glucose tolerance test**. Oral glucose tolerance test[40] was performed after 5 h fast ($n = 8$/group), (50 mg glucose, oral gavage).

**Tissue and plasma collection**. After an overnight fast, blood was collected from vena cava from terminally anaesthetized mice (PLM[3SA] and PLM[WT]) by heparinized 1 ml syringe and immediately centrifuged in pre-cooled vials (3000 rpm, 4 °C, 10 min) to obtain plasma. Skeletal muscles (gastrocnemius and soleus) were dissected and snap frozen by Wollenberger tongs for $^1$H NMR metabolic profiling. Concentrations of adiponectin, alanine aminotransferase, alkaline phosphatase, creatine kinase, free fatty acids, glucose, high density lipoprotein, insulin, lactate, lactate dehydrogenase, triacylglycerols, adrenaline and noradrenaline were measured by the Mouse Biochemistry Laboratory, Addenbrooke's Hospital, Cambridge University Hospitals NHS Trust. In a separate cohort of non-fasted terminally anaesthetized animals ($n = 5$/group), the heart, and skeletal muscle (mixed soleus and gastrocnemius) were excised, snap frozen in liquid nitrogen and stored at −80 °C for $^1$H NMR metabolomic profiling, western blotting assessment of protein expression and messenger RNA qRT- PCR. RNA was isolated using RNeasy Fibrous Tissue Kit (Qiagen) according to the manufacturer's instructions. RNA quantity and quality was assessed using Nanodrop (ThermoFisher) and only RNA with 260/280 > 1.8 was used for downstream analyses.

500 ng total RNA was reverse transcribed using Superscript VILO cDNA synthesis kit (ThermoFisher). For quantitative PCR, gene specific primer sequences (Supplementary Table 1: *glut1, glut4, cd36, caspase 3, pdk 4, ucp3, cpt1, atg3, mte-1*) were obtained from PrimerBank (https://pga.mgh.harvard.edu/primerbank/).

PCR specificity and efficiency were confirmed for each primer pair prior to use. 2.5 ng of cDNA was used per qPCR reaction in triplicates using Power SYBR Green PCR Master Mix (ThermoFisher) according to the manufacturer's instructions with the following cycling conditions: 95 C for 10 min, 40 cycles of 95 C for 15 s and 60 C for 1 min using AB 7900HT (Applied Biosystems) qPCR cycler (SDS software v 2.4).

The relative quantity of each gene was calculated using ΔΔCT method with *gapdh* as endogenous control. Myocardial protein expression of IDH3 and pyruvate dehydrogenase (PDH) was examined using manufacturer supplied methods for extraction and detection (Abcam). The following antibodies were for the western blotting experiments: rabbit-anti-IDH3A, Abcam,ab58641 (1.25 µg ml$^{-1}$), anti-rabbit, GE Healthcare NA934V (1:5000 dilution), rabbit -anti-α/β Tubulin, Cell Signalling, 2148-S / 7 (1:2000), anti-rabbit, GE Healthcare,NA934V (1:2000 dilution), mouse-anti-PDH, Abcam, ab110333 (1 µg ml$^{-1}$),anti-mouse, GE Healthcare, NA931V (1:2000 dilution). Blots were scanned and analysed using Biorad Gel 800 scanner and Image Lab software (v 6.1).

**Cardiac hypertrophy**. Myocardial hypertrophy was induced by pressure overload following supra-renal aortic constriction (banding) in 6-week-old C57BL/6J mice (20–22 g)[18]. Cardiac function and morphometry was measured in vivo in anaesthetized mice 5-weeks post-surgery using 2D echocardiography (Visualsonics Vevo 770).

**Langendorff-heart perfusions**. Mice were terminally anesthetized, hearts rapidly excised, cannulated and perfused as a standard Langendorff preparation[41] or using a custom-built NMR-compatible perfusion system in which hearts were beating spontaneously[17]. Mice were terminally anesthetized using pentobarbitone (~140 mg kg$^{-1}$ i.p.), hearts rapidly excised, cannulated and perfused as a standard Langendorff preparation[41] or using a custom-built NMR-compatible perfusion system in which hearts were beating spontaneously[17]. To study substrate preference and metabolic flux rates using $^{13}$C-labelled substrates, we adapted the standard perfusion system to utilise a counter-current membrane oxygenator[41]. At the end of each experiment, hearts were immediately freeze-clamped using Wollenberger tongs for metabolic profiling by $^1$H NMR, $^{13}$C NMR and GC-, LC-MS/MS.

**Blebbistatin and CGP 37157 Krebs-Henseleit (KH) buffer**. Pilot studies were performed in C57BL6/J perfused mouse hearts in order to determine the working concentrations of pharamacological agents used in the study: Na/K ATPase inhibitor ouabain (Sigma Aldrich,UK), myosin II inhibitor 1-phenyl-1,2,3,4-tetrahydro-4-hydroxypyrrolo[2.3-b]-7-methylquinolin-4-one (blebbistatin, Sigma Aldrich, UK) and mitochondrial Na/Ca exchanger inhibitor 7-chloro-5-(2-chlorophenyl)-1,5-dihydro-4,1-benzothiazepin-2(3H)-one (CGP13757, Sigma-Aldrich UK). In order to avoid precipitation of the blebbistatin in the vasculature and resultant ischaemia, 3.42 mmol/l blebbistatin stock solution (DMSO-diluted, aliquoted, frozen) was dissolved in KH buffer according to the previously described protocol[42]. To eliminate the possibility that any direct interaction between blebbistatin (DMSO) and ouabain mixed together in aqueous solution leads to the production of new chemical entities or binding between the two drugs, buffer samples [5 µmol/l blebbistatin (0.029% v/v DMSO), 50 µmol/l ouabain and 5 µmol/l blebbistatin (0.029% v/v DMSO) + 50 µmol/l ouabain] were analysed by mass spectrometry (Agilent 1200 LC and Agilent 6510 Q-TOF). No new chemical entities (peaks) were formed by mixing of the two pharmacological agents, thus blebbistatin did not sequester ouabain in KH. Furthermore, there was no impact of the added blebbistatin on the KH [Na$^+$], [K$^+$] and [Ca$^{2+}$] as analysed by Vetscan i-STAT1 analyser (CG8 + cartridge, Abaxis, UK).

Administration of ouabain causes immediate Na/K ATPase inhibition resulting in immediate rise in $Na_i$ accompanied by positive inotropy. The final concentration of ouabain (75 µmol l$^{-1}$) chosen for the acute $Na_i$ elevation study protocols caused significant $Na_i$ elevation which, when combined with 100 nM blebbistatin eliminated ouabain-induced inotropy (Fig. 3b).

CGP13757 was re-suspended in DMSO (1 mg/ml), aliquoted and stored at 4 °C. Final concentration used in KH (1 µmol/l) and the duration of administration was based on previously published protocols in mitochondria, cells and perfused hearts[32–34]. KH buffers containing pharmacological agents were prepared in amber glassware immediately prior to the experiment.

**Metabolic KH buffer and perfusion protocols**. For the assessment of the relative contributions of exogenous metabolic substrates (glucose and palmitate) to myocardial oxidative metabolism ($n = 7$/group) substrate-enriched metabolic KH Buffer (KH$_{metab}$) was used. In KH$_{metab}$, the glucose concentration was 5 mmol l$^{-1}$ and the following additional substrates and compounds included (in mmol/l): 1 sodium L-lactate; 0.1 sodium pyruvate; 0.5 L-glutamic acid monosodium salt monohydrate; 5 mU l$^{-1}$ insulin (NovoRapid insulin, Novo Nordisk, Denmark) and 0.3 sodium palmitate with 3% (w/v) bovine serum albumin (BSA, Proliant Biologicals, USA)[43]. Prior to inclusion, BSA was dissolved and purified as previously described[43]. KH$_{metab}$ of identical composition (in terms of components and their concentrations) but containing [U-$^{13}$C] palmitate and [1,6-$^{13}$C] glucose (Cambridge Isotopes, Goss Scientific, UK) was used for $^{13}$C NMR substrate selection/metabolic flux assessment ($^{13}$C-KH$_{metab}$).

After a 20-min functional equilibration period with KH, hearts were randomly assigned to treatment groups for perfusion:

*C57BL6/J hearts.*

(i)   KH buffer plus vehicle (0.029% v/v DMSO).
(ii)  KH buffer plus 75 µM ouabain.
(iii) KH buffer plus 100 nM blebbistatin.
(iv)  KH buffer plus 75 µM ouabain and 100 nM blebbistatin.
(v)   KH buffer plus 1 µM CGP13757.
(vi)  KH buffer plus 1 µM CGP13757, 75 µM ouabain and 100 nM blebbistatin.
(vii) Metabolic K-H plus 75 µM ouabain and 100 nM blebbistatin.

*PLM[3SA] and PLM[WT] hearts.*

i.  KH buffer perfusion, paced at 550 beats min$^{-1}$ via epicardial silver wire electrodes placed at the apex of the left ventricle and the right atrium.
ii. KH buffer perfusion unpaced.

*Banded and sham control hearts.*

i.  KH buffer perfusion.
ii. For $^{13}$C NMR substrate selection/metabolic flux analysis, PLM[3SA], PLM[WT], banded and sham control hearts were functionally equilibrated for 30 min

with KH$_{metab}$ and switched to a 40 min perfusion with $^{13}$C-KH$_{metab}$. Separate cohorts of PLM$^{3SA}$, PLM$^{WT}$, banded and sham control hearts after equilibration were perfused for 40 min with $^{13}$C- KH$_{metab}$ buffer with added 1 μmol l$^{-1}$ CGP13757.

**In situ $^{23}$Na and $^{31}$P NMR Langendorff perfusion protocols.** All in situ NMR Langendorff mouse heart perfusion experiments were carried out on a Bruker Avance III 9.4 T 400 MHz vertical wide-bore spectrometer (Bruker, Karlsruhe, Germany) equipped with triple-axis gradients, a microimaging probe and exchangeable RF coil inserts (10 mm $^{23}$Na coil or 10 mm $^{1}$H/$^{31}$P dual tune coil). Following cannulation on an MR-compatible umbilical perfusion rig, hearts were perfused with phosphate free KH buffer (11 mM glucose) and subsequently lowered into the center of the magnet for real-time multiple quantum filtered $^{23}$Na NMR (triple quantum filtered TQF and double quantum filtered DQF) quantification of Na$_i$ and $^{31}$P NMR assessment of cardiac energetics as previously described[17,41]. The hardware setup required changing between coils so that fully relaxed $^{31}$P data were acquired at baseline prior to any elevation in Na$_i$ and at the end of 30 min treatment protocol, while $^{23}$Na acquisitions were acquired throughout the functional equilibration period and during the 30 min Na$_i$ elevation/drug treatment protocols. Fully relaxed $^{31}$P experiments were acquired with a 60° flip angle, 256 scans, a repetition time of 3.8 s and a total experiment duration of 16 min. Na$_i$ quantification, the assessment of cardiac energetics and pH were performed as previously described[17,41]. MQF $^{23}$Na experiments were acquired with 192 scans, 2048 data points, sweep width of 50 ppm, an acquisition time of 200 ms, pre-scan delay of 200 ms and a total acquisition time of 1.24 min. The mixing time (τm = 3.6 ms) was calibrated for the maximumTQF signal and set to be the same for the DQF experiments. An exponential line broadening factor of 10 Hz was applied prior to Fourier transformation and subsequent baseline correction. Peak integrals were measured using Bruker Top Spin version 2.1 software.

The TQF Na$_i$ signal is a composite signal arising from the sum of various compartments each with varying electrostatic interactions, Na concentrations, and the volume occupied. The mechanism by which a TQF signal is produced is through slow rotational reorientation of the Na ion giving rise to quadrupolar relaxation when it is bound to macromolecules, and therefore largely from the intracellular compartment although there is also a contribution from the extracellular interstitial compartment. We have performed a series of in situ multiple quantum filtered $^{23}$Na NMR spectroscopy experiments in order to assess the cellular compartmentation of the $^{23}$Na TQF signal. In brief, this has involved following the TQF signal while sequentially washing out

(i)   the extracellular compartment (with a Na-free solution)
(ii)  the cytosolic compartment (with a Na-free solution plus saponin). Saponin should selectively permeabilise the cholesterol containing sarcolemma.
(iii) the mitochondrial compartment (with a Na-free solution plus Triton X-100). Triton should permeabilise all other non-cholesterol-containing membrane such as the mitochondria.

**High resolution $^{1}$H NMR of tissue extracts.** Frozen, weighed and pulverized hearts were subject to methanol/ water/ chloroform dual phase extraction adapted from Chung et al.[44] The upper aqueous phase was separated from the chloroform and protein fractions. 20–30 mg chelex-100 was added to chelate paramagnetic ions, vortexed and centrifuged at 3600 RPM for 1 min at 4 °C. The supernatant was then added to a fresh Falcon tube containing 10 μL universal pH indicator solution followed by vortexing and lyophilisation. Dual-phase-extracted metabolites were reconstituted in 600 μL deuterium oxide (containing 8 g L$^{-1}$ NaCl, 0.2 g L$^{-1}$ KCl, 1.15 g L$^{-1}$ Na$_2$HPO$_4$, 0.2 g L$^{-1}$ KH$_2$PO$_4$ and 0.0075% w/v trimethylsilyl propanoic acid (TSP)) and adjusted to pH ≈ 6.5 using 1 M hydrochloric acid and/or 1 M sodium hydroxide (<5 μL of each) prior to vortexing. The solution was transferred to a 5 mm NMR tube (Norel Inc., USA) and then analysed using a Bruker Avance III 400 MHz (9.4 T) wide-bore spectrometer (Bruker, Germany) with a high-resolution broadband spectroscopy probe at 298 K. A NOESY 1D pulse sequence was used with 128 scans, 2 dummy scans, total repetition time 6.92 s, sweep width of 14 ppm and an acquisition duration of 15 min. Data were analysed using TopSpin software version 2.1 (Bruker, Germany), FIDs were multiplied by a line broadening factor of 0.3 Hz and Fourier-transformed, phase and automatic baseline-correction were applied. Chemical shifts were normalised by setting the TSP signal to 0 ppm. Peaks of interest were initially integrated automatically using a pre-written integration region text file and then manually adjusted where required. Assignment of metabolites to their respective peaks was carried out based on previously obtained in-house data, confirmed by chemical shift, NMR spectra of standards acquired under the same conditions and confirmed using Chenomx NMR Profiler Version 8.1 (Chenomx, Canada). Peak areas were normalized to the TSP peaks and metabolite concentrations quantified per gram tissue wet weight[42,44]. Intracellular concentration of NADH, ATP + ADP, phosphocreatine, creatine, lactate, succinate, fumarate, carnitine, phosphocholine, choline, acetyl carnitine, acetate, aspartate, glutamine, glycine, alanine was analysed. The fold change with respect to the control group was then calculated for each metabolite.

**LC-MS/MS.** Lyophilised aqueous metabolite extracts were reconstituted in 350 μL ultrapure water (Millipore Corporation, USA). A series of mixed standards were prepared in ultrapure water containing 0.0025–50 μM of each metabolite. An Agilent 1100 HPLC system (Agilent Technologies, USA) consisting of an auto-sampler, a binary pump, a degasser unit and a column oven coupled to an Applied Biosystems Sciex API 3000 mass spectrometer with Turbo Ionspray interface (MDS Sciex, Canada). Chromatograpic separation was achieved using a Supelcogel C610-H column (300 mm × 7.7 mm) with a Supelcogel H guard column (50 mm × 4.6 mm) (Supelco, USA) with an isocratic flow (0.4 mL min$^{-1}$) of mobile phase consisting of 0.01% v/v formic acid and methanol (90:10) and an injection volume of 100 μL.

The HPLC eluate was split (4:1) just before the Turbo Ionspray interface resulting in a flow of 0.1 mL/min into the mass spectrometer. In order to eliminate peak to peak interference, two separate acquisitions were performed for each sample and standard. Acquisition 1 included α-ketoglutarate (145 > 101 m/z), citrate (191 > 87 m/z), isocitrate (191 > 155 m/z), fumarate (115 > 71 m/z) and lactate (89 > 43 m/z) whilst Acquisition 2 included pyruvate (87 > 43 m/z), malate (133 > 115 m/z) and succinate (117 > 73 m/z). Data were acquired using Analyst software (version 1.4.2) and metabolite concentrations in the samples were interpolated using calibration curves of each metabolite.

**GC-MS/MS.** Polar metabolites were extracted from the frozen pulverized cardiac tissue (50 mg) using the modified Folch method involving methanol water and chloroform with some modifications. Namely, a 200 μl of ice-cold distilled water with 1 mcg Norvalin as internal standard was added to the samples and 1 h sonication was performed in cold conditions. This was followed by addition of 500 μl HPLC grade methanol (ice cold) to each samples with 1 h sonication in ice cold conditions. Subsequently, the methanol: water extract was transferred using glass Pasteur pipette to a new labelled high grade Eppendorf tube and 500 μl chloroform was added to each tube, vortexed for 1 min followed by 15 min shaking on the shaker at high speed. Subsequently, the Eppendorf tubes were centrifuged at 13,000 rpm, 4 C for 15 min and the top polar layer was aspirated to a clean Eppendorf tubes. The polar extract was dried using a speedvac and stored in −80 freezer for subsequent derivatization.

**Derivatization method.** All derivatization steps were carried out in a fume hood. In order to derivatize proteinogenic amino acids, organic acids and glycolytic intermediates for GC-MS analysis, the dried extract was incubated at 95 °C in open tubes in order to remove any residual moisture in the samples. The dried extract was solubilized in 40 μl of 2% methoxyamine HCL in pyridine (Sigma-Aldrich, Dorset,UK) followed by 60 min incubation at 60 °C and subsequently 60 μl N-tertbutyldimethylsilyl-N-methyltrifluoroacetamide (MTBSTFA) with 1% (w/v) tertbutyldimethyl-chlorosilane (TBDMSCI) (Sigma-Aldrich, Dorset, UK) derivatization reagent was added. The suspension was incubated for an hour at 60 °C in a well-sealed tube to prevent evaporation. Finally the samples were centrifuged at 13,000 rpm for 5 min and the clear supernatant was transferred to a chromatography vial with a glass insert (Thermo Fisher, Scientific, Chromacol, Hertfordshire, UK) and proceeded immediately to GC-MS analysis.

**GC-MS/MS analysis.** For analysis of the derivatized samples an Agilent 7890B Series GC/MSD gas chromatograph with a polydimethylsiloxane GC column coupled, with a mass spectrometer (GC-MS) (Agilent Technologies UK Limited, Stockport, UK) was used. Prior to sample analysis the GC-MS was tuned to a full width at half maximum (FWHM) peak width of 0.60 a.m.u. in the mass range of 50 to 650 mass to charge ratio (m/z) using PFTBA tuning solution.

One microlitre of sample was injected into the GC-MS in splitless mode with helium carrier gas at a rate of 1.0 ml min$^{-1}$. The inlet liner containing glass wool was set to a temperature of 270 °C. Oven temperature was set to 100 °C for 1 min before ramping to 280 °C at a rate of 5 °C min$^{-1}$. Temperature was further ramped to 320 °C at a rate of 10 °C min$^{-1}$ held at 320 °C for 5 mins. Compound detection was carried out in full scan mode in the mass range 50–650 m/z, with 2–4 scans s$^{-1}$, a source temperature of 250 °C, a transfer line temperature of 280 °C and a solvent delay time of 6.5 min. The injector needle was cleaned with acetonitrile three times before measurement commencement and three times following every measurement thereafter. The raw GC-MS data were converted to common data format (CDF) using the acquisition software and further processing of the isotope data including isotope correction and mass isotopomer analysis /batch quantification was performed on metabolite detector software. To determine absolute concentration, a 7 point calibration series covering the mass range of 0–8.46 μM was prepared in triplicates with 100 μl of 8.5 μM of internal standard added to each sample of the calibration series and were extracted as the method outlined above. The dried extract were then derivatised followed by GCMS analysis. For absolute quantification, the ratio of peak area of each concentration to the peak area of internal standard was calculated and plotted against the ratio of the concentration of analyte with respect to the concentration of internal standard to generate the equation and estimate the linear dynamic range. Subsequently, the raw peak area for each analyte of interest was calculated using the metabolite detector software followed by normalizing the response to the internal standard peak area.

**Tissue extraction for $^{13}$C NMR**. Frozen hearts were weighed and ground to a fine powder under liquid nitrogen and extracted at 4 °C with 6% perchloric acid (PCA) in a ration 5:1. The suspension was centrifuged at 4000 RPM, 4 °C for 10 min and a known volume of supernatant decanted and neutralised with 6 M KOH to pH7.0 at 4 °C. The mixture was centrifuged and the supernatant lyophilised at −40 °C. Lyophilised tissue extracts were reconstituted in 0.6 ml of 50 mM deuterated phosphate ($KH_2PO_4$) buffer pH 7.0 lyophilised and resuspended in $D_2O$. A small amount of chelating resin (Chelex-100) was added to samples to remove any paramagnetic ions and filtered through a 0.22 μm syringe filter into 3 mm NMR tube.

**High resolution $^{13}$C NMR of tissue extracts**. High-resolution $^1$H-decoupled $^{13}$C NMR spectra were acquired under automation at 298 K on a Bruker Avance III 700 (16.4 T) NMR spectrometer (Bruker Biospin, Coventry, UK) equipped with a 5 mm TCI helium-cooled cryoprobe and a refrigerated SampleJet sample changer. The temperature was allowed to stabilise for 3 min after insertion into the magnet. Tuning, matching and shimming was performed automatically for each sample and the $^1$H pulse length was calibrated on each sample and was typically around 8 μs. 1D $^1$H-decoupled $^{13}$C spectra (zgpg60) were acquired with 8192 transients, a spectral width of 200 ppm, 64 K data points, a mixing time of 10 ms, relaxation delay of 1 s and repetition time of 2 s. 1H-decoupling was achieved using a WALTZ65 sequence during the relaxation delay and acquisition. Spectra were processed in the manufacturer's software (Topspin 3.2.6). Free induction decays were multiplied with an exponential function (line broadening of 0.25 Hz), Fourier transformed, phase correction was performed manually and automatic baseline correction was applied. Representative spectra are shown in Fig. 1b. The relative contributions of exogenous $^{13}$C substrates (palmitate vs glucose) to oxidative phosphorylation were determined from $^{13}$C glutamate isotopomer labelling patterns (Fig. 1b) using tcaCALC$^{tm}$ software (v2.07)[22,24].

**In silico modelling**. In silico simulations were performed using the metabolic network of the cardiomyocyte CardioNet[21,22,45]. Mathematical modelling has previously been used to study the dynamics of cardiac metabolism in response to stress[20,21,46], and CardioNet has been successfully applied to identify limiting metabolic processes and estimate flux distributions[20,22]. Flux balance analysis (FBA) allows to estimate flux rates in a cellular model based on metabolic constraints that are defined by the extracellular environment (e.g. oxygen and nutrient supply), cellular demands (e.g. proliferation, contraction) and tissue type (e.g. heart vs. liver). This modelling approach combines biochemical network models with optimality problems, which describe different cost or benefit functions and allow us to include experimental data, for example metabolite levels, enzyme levels or flux rates.

The advantage of flux balance analysis is that it considers system-wide effects of processes and allows us to assess metabolic limitations in an unbiased approach. We applied flux balance analysis to identify which reactions are involved in myocardial metabolic adaptations to high levels of $Na_i$.

**Mathematical modelling of myocardial metabolic adaptations to $Na_i$ elevation**. Metabolic flux distributions were calculated using constrained based modelling. To calculate flux rate changes ($v_i$), we constrained the model for each metabolite using experimentally determined metabolite concentrations ($^1$H NMR, LC-MS/MS, effluent lactate production, $^{13}$C substrate utilization measurements) to maximize cardiac work reflected by ATP hydrolysis ($v_{ATPase}$). The decision to optimize steady-state ATP production was not arbitrary. Our data show that while Na elevation re-programs metabolism, it does so while not compromising energetics—as demonstrated by our $^{31}$P-NMR measurements showing maintained ATP, PCr, PCr/ATP ratios and pH$_i$. ATP and PCr concentrations are clearly a product of both production and consumption, however, our wet-biology experimental design tries to keep consumption as constant as possible (by the titration of contractility with blebbistatin) thereby keeping steady-state ATP consumption constant. Simulations were run with boundary conditions reflecting the metabolite composition of the perfusion buffer and experimentally measured uptake and release rates of substrates. At the same time, various metabolites, including amino acids and lipids, were set to previously reported values[21,47–49], in order to mimic the ex vivo experimental conditions. Based on these constrains we first determined flux distributions ($v_m$) under baseline control conditions. We then calculated fold-changes (FC) for experimentally measured metabolite concentrations between baseline controls and treatment groups (acute and chronic $Na_i$ elevation), and used these FC to further constrain fluxes ($v_m$) for the synthesis and/or degradation of intracellular metabolites. We included FC based on the assumption that changes in metabolite concentrations under experimental conditions are accompanied by a proportional increase or decrease in the respective flux for the metabolite pool. By using metabolite level changes (fold changes) to estimate flux rate changes ($v_{FC}$), we imply that the altered steady-state concentrations of metabolites are reflected in the newly evolved flux state and potentially limit metabolic functions.

The following flux balance analysis was applied to identify steady-state flux distributions that are in agreement with applied substrate uptake and release rates, and changes in metabolite pools:

$$\max v_{ATPase} \tag{1}$$

subject to (1)

$$S \cdot v = 0, \tag{2}$$

$$v_i^{(-)} \leq v_i \leq v_i^{(+)}, \tag{3}$$

$$L_j^{(-)} \leq v_j \leq L_j^{(+)} (j = j_1, j_2, \dots), \tag{4}$$

$$v_m \leq FC_m \cdot v_m^0 (m = m_1, m_2, \dots), \tag{5}$$

where $v_i$ denotes the flux rate change through reaction $i$, $v_j$ denotes the measured uptake or secretion rate through reaction $j$, $S$ is the stoichiometric matrix, and $v_i^{(-)}$ and $v_i^{(+)}$ are flux constraints. The CPLEX LP solver was used to find the solution to the FBA problems. The logarithm of the metabolic flux rate values is presented in the form of heatmaps. Metabolic reactions are clustered according to their association to metabolic pathways and plot colours indicate estimated flux rates for each metabolic reaction. All reactions and their metabolic subsystems, classified in the Kyoto Encyclopedia of Genes and Genomes database[50].

**Data statistics and reproducibility**. Data are presented as mean ± SEM and analysed blind to phenotype or treatment. Statistical analysis was conducted using GraphPad Prism (v 8.3) and Microsoft Excel (v.16.16.15). All data were obtained from a minimum of two independent experiments. Comparison between groups was by Student's $t$-test (Gaussian data distribution), two-way analysis of variance (ANOVA) with Bonferroni's correction for multiple comparison and one-way ANOVA using Bonferroni's correction for multiple comparisons where applicable. After pharmacological agent treatment (ouabain, blebbistatin, CGP13757), hearts were compared to baseline control values for the same genotype. Metabolite fold changes of the ratio of treated (T) vs control (C) groups were calculated and the fold change. Propagated standard error (SEM) of the ratio was calculated using the formula $SEM_{(T/C)} = (T/C)\sqrt{(SEM_T/T)^2 + (SEM_C/C)^2}$, assuming the covariance between the two groups is zero, i.e., C and T are uncorrelated. Metabolic differences between the flux distributions were analysed by R Statistics (version 1.2.1335 for Fedora/RatHat 7 64-bit, R version 3.0.1, Boston Massachusetts, USA, www.rstudio.com)[51]. Datasets were tested for normal distribution using the Shapiro–Wilk test. Groups were compared using non-parametric (Kruskal–Wallis) test methods. Unsupervised hierarchical clustering and PCA were conducted using R-Studio. Heat maps and z-scores were generated using R-Studio with the heatmap.2 function and viridis colour palettes from the R-package gplots (version 3.0.1.1)

Z-scores were calculated as follows:

$$z = \frac{(X - \mu)}{\sigma} \tag{6}$$

where $z$ denotes the z-score, $X$ is the average flux rate for a given reaction within an experimental group (control, acute and chronic Na elevation), $\mu$ is the population mean, $\sigma$ and is the standard deviation. Z-scores were calculated for each reaction (each row in the heat map) including all control, acute and chronic Na elevation values. Each calculated z-score was assigned a colour as depicted in the heat map. The similarity between groups was assessed using a Euclidean distance and the number of clusters was determined using the $k$-means algorithm. We applied the Elbow method to determine optimal numbers of clusters. Columns (experimental groups) were clustered hierarchically according to dissimilarities between clusters with the squared Euclidean distances between cluster means calculated.

Differences were considered significant when $P < 0.05$.

**Reporting summary**. Further information on research design is available in the Nature Research Reporting Summary linked to this article.

## Data availability

All data supporting the results presented herein are available from the corresponding author upon reasonable request. The source data for all the graphs and uncropped blots in the main figures and supplementary information are provided as a Source File (Source Data.xcl). Cardionet modelling data are provided in a Supplementary File 1. Database availability: Cardionet MODEL1212040000; Kegg database: https://www.genome.jp; Gene specific primer sequences are available from PrimerBank. All metabolomic data are archived with DataDryad. Source data are provided with this paper.

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

## Acknowledgements

We acknowledge A. McNeilly, R. McCrimmon, M. Kerr, L. Heather, C. Imberti for technical assistance, Merck Sharpe and Dohme UK (S. Strickfuss, S. Hamilton) for access to LC-MS/MS equipment, A. Atkinson and the Centre for Biomolecular Spectroscopy, King's College London (KCL) for access to the 700 MHz NMR. This work was supported by a British Heart Foundation (BHF) Programme Grant (RG/12/4/29426) (M.J.S. and W. F.), KCL BHF Centre of Research Excellence (RE/08/003), the Friede Springer Herz Stiftung (A.K.), the Roderick McDonald Research Fund (15RDM005 to A.K.), the American Heart Association (17POST33660221 to A.K.), NIH (K99-HL141702 to A.K., R01-HL-61483 to H.T.); BHF Intermediate Basic Science Fellowship (FS/16/21/31860) (SE); NIHR Biomedical Research Centre at Guy's and St Thomas' NHS Foundation Trust and KCL; the Centre of Excellence in Medical Engineering funded by the Wellcome Trust and Engineering and Physical Sciences Research Council (EPSRC) (WT 088641/Z/ 09/Z); KCL Comprehensive Cancer Imaging Centre funded by the Cancer Research UK (CRUK) and EPSRC in association with the Medical Research Council (MRC) and the Department of health (DoH). The views expressed are those of the authors and not necessarily those of the NHS, the NIHR or DoH. TRE is grateful for support from CRUK

and EPSRC in association with MRC and DoH (C1060/A10334). We thank Sofia and Stefan Ricciarelli for timing their arrivals so the experiments could be completed.

## Ethics declaration

This investigation was approved by King's College London Ethical Review Committee and conforms to UK Home Office Guidance on the Operation of the Animals (Scientific Procedures) Act, 1986 (Home Office Project Licences MJS:PF75E5F7F and P856ECBBE).

## Author contributions

D.A.: study design, data acquisition, analysis and interpretation: ex vivo and in situ (NMR) Langendorff-heart perfusions, $^{23}$Na, $^{1}$H, $^{31}$P, $^{13}$C NMR spectroscopy, in vivo metabolic profiling. A.K.: CardioNet in silico modelling. MB: TAC surgery and in vivo cardiac function assessment. B.O.B.: LC MS/MS analysis, blebbistatin experimental design. D.S.T.: western blotting and real-time PCR analysis. S.E.: real-time PCR analysis, experimental design. A.T., D.T.: LC Ms/Ms analysis, W.F., M.J.S.: obtained funding, T.R.E.: NMR spectroscopy expertise ($^{23}$Na, $^{31}$P, $^{1}$H): experimental design, data acquisition, data analysis, study design and interpretation. M.J.S.: study design and interpretation. D.A., T.R.E., A.K., H.T., M.J.S.: wrote and edited the MS.

## Competing interests

The authors declare no competing interests.
