## [Peer Review File · Nature Communications]

Reviewers' comments:

Reviewer #1 (Remarks to the Author):

This is an interesting and very creative series of experiments addressing the linkage between changes in cytosolic Na and multiple downstream consequences for high-energy phosphate metabolism and substrate oxidation pathways. There are many strengths, for example including ^{31}P NMR which provides assurance that the tissues are not ischemic, or the design of experiments that separates contractile changes from changes in intracellular sodium. The work would have been strengthened if oxygen consumption was measured to allow calculation of fluxes in the various pathways. The key issue here is whether elevated intracellular Na causes the changes in metabolism that are observed in hypertrophied hearts, certainly of fundamental importance. A few questions:

1. The ^{23}Na NMR multiple quantum filtering is detecting intracellular sodium. Presumably the biexponential decay is due to interaction of sodium with electrostatic sites on ordered biopolymers in the cell. Since the physical environment of the mitochondrial matrix is different from that of the cytosol, is sodium from the matrix over-represented or under-represented in the MQ data? I did not see the actual results of the ^{23}Na experiments.

2. The preferential oxidation of glucose in hypertrophy is attributed in this paper to an increase in intracellular Na and therefore a decrease in mitochondrial calcium. But it is not clear how the investigators envision the link to metabolism. To oversimplify, the increase in cytosolic Na presumably decreases intra-mitochondrial calcium, which would reduce activity of pyruvate dehydrogenase phosphatase, in turn reducing PDH flux and glucose oxidation, just the opposite of what was reported here and in the literature. Of course glucose metabolism is highly regulated at many other steps, but the proposed cascade of events occurring between changes in sodium in the cytosol, and effects on glucose or fatty acid oxidation was not quite clear.

3. The comments on lines 123-130 are confusing to me. The classical teaching is that cardiac glycosides such as ouabain, exactly as outlined later in the paper, cause increased contractility mediated by inhibition of the Na-K ATPase, an increase of $[\text{Na}^+]_{\text{cytosol}}$ and an increase of $[\text{Ca}^{++}]_{\text{cytosol}}$ because of inhibition of calcium export via the sodium-calcium exchanger in the sarcolemma. If $[\text{Na}^+]_{\text{cytosol}}$ increases, then the sodium – calcium exchanger on the inner mitochondrial membrane causes the export of mitochondrial Ca^{++} and therefore reduced intramitochondrial $[\text{Ca}^{++}]$, thereby inhibiting the various dehydrogenases, as outlined. Therefore the rate of NADH production and ATP production must be reduced as suggested by the authors. Yet increased developed pressure must be supported by increased ATP production, so it would help me if these relationships could be clarified.

Minor:

Line 132 in supplemental, Bruker 9.4 (not 9.7)

Figure 4. Many of the components in the figure are whited out by overlying boxes.

Reviewer #2 (Remarks to the Author):

The manuscript by Aksentijević et al seeks to test the hypothesis that elevated intracellular myocardial sodium (NAi) may be a unifying mechanism to explain the timing, energetic, and functional dysfunction in various models. Specifically, this was tested in a mutant phospholemman transgenic mouse (PLM3SA) which leads to chronic elevated NAi, pharmacological (i.e. ouabain) acute elevations of NAi, or surgical cardiac hypertrophy (TAC). The metabolism was assessed by a combination of NMR, MS, and in silico analysis. The authors found that in all three treatment groups, fatty acid utilization was decreased and in both the PLM3SA TG and TAC models glucose utilization was increased. Furthermore, chemical inhibition of the Na(+)/Ca(2+) exchanger (NCLX) with CGP37157 mostly attenuated this metabolic switch. These changes were accompanied by depletion of a number of metabolic intermediates in PLM3SA mice but an increase or no change in the current severity and duration of TAC. Furthermore, in silico modeling mirrored their predictions (e.g. general decrease in TCA, except citrate synthase with a general decrease in FAO). Consistent with some hypotheses about these metabolic changes there was a predicted increase in OXPHOS linked to a change in the regulation of ROS. Together these studies support the long-standing metabolic switch known to occur in the heart in response to certain chronic stresses. It also now links NAi levels to these regulations and suggests, that at least in the compensated pre-failure state, the changes may be adaptive. However, as they are maintained they are insufficient to compensate. Importantly, these studies highlight the timing of the metabolic changes in cardiac hypertrophy and suggest that they precede, and may be independent, of the structural and functional remodeling. Overall, this is a succinct but well-designed study adding yet another important piece to our understanding of the regulation of cardiac metabolism. I only have a few minor concerns that if addressed will further improve the potential impact of the study.

1. In the text, the authors state that there is not an overt systemic metabolic phenotype. This is mostly true and the data do strongly support the statement. It is interesting to note the statistically

significant elevation in serum glucose and insulin (Table S2). The OGTT (Fig. S1A) and other data suggest this is a minor contributing factor. Please comment on this in the discussion.

2. Some of the figures are hard to interpret. The inclusion of color helps some, but they still run together (e.g. Fig. 1G). It would help to spread the individual metabolites to include white space (there is plenty available to the right as the figure is currently laid out) and/or add dotted lines between them. Figure 2D is slightly better in this regard.

Reviewer #3 (Remarks to the Author):

This is a nice study showing that acute elevation of intracellular $[Na]_i$ in cardiac myocytes causes a metabolic shift (independent of increased work) quite similar to some known to occur in heart failure (substrate shift from fatty acids toward glucose utilization). The aspect of the results that is especially remarkable is that even the acute Na/K-ATPase (NKA) inhibition causes this shift in minutes, and that the effects are similar to those seen in a chronic phospholemman (PLM) knockin mouse that inhibits NKA and raises $[Na]_i$ and also a pressure-overload model of cardiac hypertrophy. These metabolic changes have usually been assumed to require transcriptional regulation and altered gene expression, but this study suggests that they are acutely caused by elevated $[Na]_i$ (and presumably reduced $[Ca]_{mito}$) before major gene transcription effects are expected. I do have a number of suggestions that could strengthen the paper and make the conclusion more convincing.

Main points:

1. The most exciting aspect of this study is the possibility that these metabolic shifts might occur acutely by direct consequences of $[Na]_i$ and possibly mitochondrial $[Ca]$ via acute regulation of the existing protein machinery, before substantial changes in gene regulation and consequent protein expression could have occurred. To strengthen this conclusion, it would be valuable to show that:

a) myocyte protein expression (with respect to a panel of key proteins) does not change appreciably in the 30 min ouabain-Blebbistatin case or CGP exposure vs. control,

b) another acute mechanism that should lower mitochondrial $[Ca]$ has similar acute effects on metabolism (e.g. Ru360) as Ouabain-Blebbi,

c) $[Ca]_{mito}$ is actually reduced by acute $[Na]_i$ elevation and reversed by CGP.

d) the computational model explains this mechanistically by simple $[Ca]_{mito}$ and/or $[Na]_i$ changes. That might have been the aim, but as written it totally misses the mark (see below).

2. Ln 179-182. In Fig 1 CGP alone seemed to cause a slight downward shift in the FA/Glucose utilisation, compared to control. While the individual effects (FA or Glu) were not significantly different, this ratio seems to go in the opposite direction to that expected from the main thesis. And in line 186-7 there do seem to be some CGP effects on metabolites vs. WT.

3. It seems important that CGP effects in Fig 2 (like ouabain effects later) were manifest in 30 min of exposure. This relates to point #1 above, and may be worth highlighting in that context (acute regulation vs. transcription).

4. The CardioNet computer model and its use are very poorly described and hence difficult to evaluate. The reference in the main manuscript (24) does not even mention CardioNet (although O'Rourke & Maack do mention the Cortassa et al. model), nor are the different references in the Supplement (10,11) primary papers that developed CardioNet. There is no acknowledgement as to the strengths or weaknesses of the model, or its precision with respect to mouse ventricular myocytes. It also sounds like various things in the model were adjusted, but the reasoning or justification is not clear (and that should be distinguished clearly from the use of the model). If you manipulate the model to fit your data, you can no longer use it as a "logical" test of plausibility. It seems to me that the main value of the model in this study would be to test whether the simple change in $[Na]_i$ or $[Ca]_{Mito}$ by itself (e.g. by known effects on dehydrogenases) would be expected based on the existent model to cause the metabolic changes observed. The only things changed should be either $[Na]_i$, $[Ca]_{Mito}$ or if CardioNet does not have good mitochondrial Na-Ca flux descriptions you could change activity of the specific Ca-dependent processes. Maybe that is what you are doing (Ln 258, 70, 80 and 90%) but even that is unclear. On line 262-3 a 0.7 fold decrease in flux is stated; is this a reduction to 70% of V_{max} (30% remaining) or a reduction to 70% V_{max} . Also flux should be an output that depends not only on V_{max} , but also levels of substrates and products. This section really needs a major revision if it is to be useful. Ln 323-326 should be more explicit about the modeling as a plausibility test of the Na-Ca hypothesis you propose.

5. The CGP37157 concentration and its expected (or measured) level of NCLX inhibition should be stated. It can only be partial inhibition, because as we now know complete NCLX knockout is lethal (PMID: 28445457). See also point 1c above.

Minor:

Pg 1. Author affiliations for ALL authors should be indicated (by superscript).

Pg 2. Last line in Abstract should be "...be a new approach..."

Pg 8: It may be worth indicating that the rise of $[Na]_i$ in heart failure may be due in part to elevated late Na current (many refs), diastolic Na influx via Na channels (Despa et al. 2002) and/or reduced NKA function (author's work?).

Reviewers' comments:

Reviewer #1 (Remarks to the Author):

This is an interesting and very creative series of experiments addressing the linkage between changes in cytosolic Na and multiple downstream consequences for high-energy phosphate metabolism and substrate oxidation pathways. There are many strengths, for example including ^{31}P NMR which provides assurance that the tissues are not ischemic, or the design of experiments that separates contractile changes from changes in intracellular sodium. The work would have been strengthened if oxygen consumption was measured to allow calculation of fluxes in the various pathways. The key issue here is whether elevated intracellular Na causes the changes in metabolism that are observed in hypertrophied hearts, certainly of fundamental importance.

We thank the Reviewer for their comments and for finding our study interesting and creative. We have taken Reviewers' comments onboard and have made amendments to our manuscript (changes in red font).

A few questions:

1. The ^{23}Na NMR multiple quantum filtering is detecting intracardiac sodium. Presumably the biexponential decay is due to interaction of sodium with electrostatic sites on ordered biopolymers in the cell. Since the physical environment of the mitochondrial matrix is different from that of the cytosol, is sodium from the matrix over-represented or under-represented in the MQ data? I did not see the actual results of the ^{23}Na experiments.

We thank the Reviewer for this point. This is indeed correct – the TQF signal is a composite signal arising from the sum of various compartments each with varying electrostatic interactions, Na concentrations, and the volume occupied. The mechanism by which a TQF signal is produced is through slow rotational reorientation of the Na ion giving rise to quadrupolar relaxation when it is bound to macromolecules, and therefore largely from the intracellular compartment although there is also a contribution from the extracellular interstitial compartment. We have now performed a series of in situ multiple quantum filtered ^{23}Na NMR spectroscopy experiments in order to assess the cellular compartmentation of the ^{23}Na TQF signal. Briefly, this has involved following the TQF signal while sequentially washing out -

1. the extracellular compartment (with a Na-free solution)
2. the cytosolic compartment (with a Na-free solution plus saponin). Saponin should selectively permeabilise the cholesterol containing sarcolemma.
3. the mitochondrial compartment (with a Na-free solution plus Triton X-100). Triton should permeabilise all other non-cholesterol-containing membrane such as the mitochondria.

The following figure shows the result of these experiments -

A

B

Figure 1: Triple Quantum Filtered ^{23}Na NMR spectroscopy assessment of the Intracellular Na compartmentation in perfused heart (n=5 Wistar rat hearts)

Panel A shows a water-fall plot of a single experiment and panel B shows the data averaged from 5 separate hearts.

Note: we tried initially to do this experiment in a different way. In these initial experiments (not shown) we used a Na-containing perfusate which included a shift reagent (TmDOTP) and then sequentially permeabilised the cell – first with saponin and then Triton. Theoretically, as the TmDOTP sequentially accesses each compartment this should sequentially shift the Na signal from the extracellular, intracellular and mitochondrial compartments along the spectral axis as they became sequentially permeabilised. Unfortunately, it transpired that TmDOTP and saponin interact rendering the spectra uninterpretable. Hence, we chose to use the above Na-free washout protocol in the absence of shift reagent.

In Panel B, the open circle symbol represents the fraction of the TQF signal that is attributable to the intracellular compartment estimated in previous TmDOTP experiments (see Eykyn et al 2015).

Briefly, our interpretation of the data in the figure above is as follows -

1. When the heart is exposed to a Na-free solution the washout of the vascular and extracellular space is very rapid (we have previously estimated this to be <40 seconds in isolated hearts using flame photometry). After this, there is a steady fall in the TQF signal as Na leaks out of the cell and intracellular Na falls. **Note:** Previous studies using ion-selective microelectrodes (see Ellis 1977) have reported that intracellular Na falls with a very rapid initial component. For example, in the experiments of Ellis (1977), in what were likely to be relatively poorly superfused Purkinje fibres, changing extracellular Na from 140 to 14mM caused a rapid fall in intracellular Na from 10 to 2.9mM with 90% of this change complete in 3.2 mins. This is very similar to the time course of intracellular Na decline we see in our experiments when switching to a Na-free solution.
2. On changing to the Na-free saponin-containing solution, the TQF signal continues to fall as the membrane is permeabilised. We were slightly surprised that saponin did not accelerate significantly the rate of fall of intracellular Na - suggesting that the rate of efflux before saponin was not significantly different to that after saponin. However, the TQF signal then fell to a plateau at about 9% of the baseline.
3. On exposure to Triton X-100, the remaining TQF signal was eliminated – presumably as the mitochondria were permeabilised.

On the basis of the data shown above, we therefore estimate the following compartmental contributions to the total TQF signal –

Vascular + extracellular space = 47%

Intracellular signal = 53%

Cytoplasmic signal = 44%

Mitochondrial signal = 9%

In the experiments reported in the paper, we assume that the total Na in the vascular and extracellular compartments does not change acutely and simply provides an offset to the intracellular signal. In order to control for this eventuality, however, we measure the DQF signal. As we have previously shown (Eykyn et al 2015), the DQF signal entirely originates from the vascular and extracellular unbound Na. So, we use this DQF signal as an internal control to verify that this compartment is not changing and any changes in TQF are thus attributable to changes in the intracellular compartment. It is interesting to note that while the mitochondria occupy about 40% of the cell volume, and have a similar internal Na concentration to that in the cytosol, they

contribute only 20% of the intracellular TQF signal suggesting that Na electrostatic buffering in the mitochondria differs from that in the cytoplasm.

This figure and a description of the method used and the results are now included in the data supplement (Supplementary Figure 1) and briefly referred to in the revised manuscript (lines 452-455).

2. The preferential oxidation of glucose in hypertrophy is attributed in this paper to an increase in intracellular Na and therefore a decrease in mitochondrial calcium. But it is not clear how the investigators envision the link to metabolism. To oversimplify, the increase in cytosolic Na presumably decreases intra-mitochondrial calcium, which would reduce activity of pyruvate dehydrogenase phosphatase, in turn reducing PDH flux and glucose oxidation, just the opposite of what was reported here and in the literature. Of course glucose metabolism is highly regulated at many other steps, but the proposed cascade of events occurring between changes in sodium in the cytosol, and effects on glucose or fatty acid oxidation was not quite clear.

We thank the reviewer for this point. We agree with this assessment. The crucial ATP supply-demand relationship has been proposed to be affected by elevated cytosolic Na_i , in agreement with previously published work by Iwai et al. (2002). When calcium-sensitive dehydrogenases were inhibited as per the supplemental modelling figure, the flux through the TCA cycle indeed goes down as previously suggested in experiments using isolated organelles and cells. However, our experiments in perfused hearts as well as Cardionet model constrained to the experimental data have shown that the metabolic flux is maintained or if anything goes up to maintain energetics constant but at the expense of increased utilization of anaplerotic substrates. What we are proposing is not that oxidative metabolism shuts down but that the impact on the Ca-dependent enzymes 're-routes' substrate metabolism such that anaplerotic and other substrate pathways become more important. This leads to **metabolic inefficiency** rather than impaired energetics. Thus our data suggest that supply-demand is not mismatched but adapted to compensate.

Specifically we show that the heart switches to using other substrates (including amino acids) and produces lactate which can in turn be used as metabolic fuel via intracellular lactate shuttle (Brooks et al 1999). Our findings suggest that with the contribution of fatty acid β -oxidation to ATP provision decreased (due to drop in mitochondrial Ca), ATP provision for contractile work is maintained due to compensatory increase of amino acid and carbohydrate utilisation (pyruvate, lactate) to replenish Krebs cycle intermediates. Maintained contractile function and ATP provision in the face of high Na_i comes at the cost of increased exogenous substrate utilization and enhanced fluxes leading to significant metabolite depletion. These findings are in keeping with previously reported features of metabolic inefficiency arising from simultaneously increased yet mismatched metabolic fluxes (including high turnover of ATP) and a failure of fatty acid β -oxidation to compensate (Masoud et al 2014). In addition our work is in agreement with previously reported tightly coupled ATP supply for the sodium pump arising preferentially from carbohydrate metabolism (Okamoto et al 2001, Cross et al 1995). This is highlighted in our discussion (lines 306-318, 329-337, 339-357).

3. The comments on lines 123-130 are confusing to me. The classical teaching is that cardiac glycosides such as ouabain, exactly as outlined later in the paper, cause increased contractility mediated by inhibition of the Na-K ATPase, an increase of [Na⁺]_{cytosol} and an increase of [Ca⁺⁺]_{cytosol} because of inhibition of calcium export via the sodium-calcium exchanger in the sarcolemma. If [Na⁺]_{cytosol} increases, then the sodium – calcium exchanger on the inner mitochondrial membrane causes the export of mitochondrial Ca⁺⁺ and therefore reduced intramitochondrial [Ca⁺⁺], thereby inhibiting the various dehydrogenases, as outlined. Therefore the rate of NADH production and ATP production must be reduced as suggested by the authors. Yet increased developed pressure must be supported by increased ATP production, so it would help me if these relationships could be clarified.

We thank the reviewer for raising this point. This question neatly summarises the conundrum. Indeed it is this mismatch between energy demand and energy supply that must drive the metabolic reprogramming. If the heart is unable to meet energy demand by stepping up mitochondrial oxidative metabolism (as it would normally do on a rise in cytosolic Ca), then it has to alter its substrate utilisation as well as altering glycolytic metabolism. This is what appears to happen – the heart switches to using other substrates. However, in our study, the PLM^{3SA} and ouabain+ blebbistatin protocol hearts do not have increased developed pressure thus ATP demand (and by extension ATP supply) and overall energetics are unchanged compared to baseline controls. Yet we still observed a rerouting of metabolic substrates under these conditions suggesting an adaptive response to the elevation in Na_i.

In a 'healthy' heart perfused with ouabain + blebbi this reprogramming does not compromise energetics (ie ATP/PCr), however, a failing heart may not be so accommodating. Increased developed pressure in a healthy heart induced by beta adrenoceptor stimulation will be associated with increased Na extrusion as a consequence of PLM-Na/K ATPase stimulation and hence Na will not be elevated in normal chronotropy/inotropy (Despa et al 2005). In a normal healthy heart both increased chronotropy and increased inotropy are associated with an increase in the time-averaged Ca which will activate mitochondrial enzymes to step-up ATP production. It is only when cytoplasmic Na is elevated that this feed-forward control goes wrong. This is supported by our present paper and the work of O'Rourke et al and Maack et al.

Minor:

Line 132 in supplemental, Bruker 9.4 (not 9.7)

This has now been corrected. (Supplementary Material line 128).

Figure 4. Many of the components in the figure are whited out by overlying boxes.

This editing lapsus has now been corrected-issue arose during conversion to pdf for submission.

Reviewer #2 (Remarks to the Author):

The manuscript by Aksentijević et al seeks to test the hypothesis that elevated intracellular myocardial sodium (NAi) may be a unifying mechanism to explain the timing, energetic, and functional dysfunction in various models. Specifically, this was tested in a mutant phospholemman transgenic mouse (PLM3SA) which leads to chronic elevated NAi, pharmacological (i.e. ouabain) acute elevations of NAi, or surgical cardiac hypertrophy (TAC). The metabolism was assessed by a combination of NMR, MS, and in silico analysis. The authors found that in all three treatment groups, fatty acid utilization was decreased and in both the PLM3SA TG and TAC models glucose utilization was increased. Furthermore, chemical inhibition of the Na(+)/Ca(2+) exchanger (NCLX) with CGP37157 mostly attenuated this metabolic switch. These changes were accompanied by depletion of a number of metabolic intermediates in PLM3SA mice but an increase or no change in the current severity and duration of TAC. Furthermore, in silico modeling mirrored their predictions (e.g. general decrease in TCA, except citrate synthase with a general decrease in FAO). Consistent with some hypotheses about these metabolic changes there was a predicted increase in OXPHOS linked to a change in the regulation of ROS. Together these studies support the long-standing metabolic switch known to occur in the heart in response to certain chronic stresses. It also now links NAi levels to these regulations and suggests, that at least in the compensated pre-failure state, the changes may be adaptive. However, as they are maintained they are insufficient to compensate. Importantly, these studies highlight the timing of the metabolic changes in cardiac hypertrophy and suggest that they precede, and may be independent, of the structural and functional remodeling. Overall, this is a succinct but well-designed study adding yet another important piece to our understanding of the regulation of cardiac metabolism. I only have a few minor concerns that if addressed will further improve the potential impact of the study.

We thank the Reviewer for the thorough critique of our work and for finding our study well-designed.

1. In the text, the authors state that there is not an overt systemic metabolic phenotype. This is mostly true and the data do strongly support the statement. It is interesting to note the statistically significant elevation in serum glucose and insulin (Table S2). The OGTT (Fig. S1A) and other data suggest this is a minor contributing factor. Please comment on this in the discussion.

We thank the Reviewer for this point. Despite a small alteration in serum glucose and insulin concentration in PLM^{3SA} mice (Supplementary Table 2), there was no functional consequence as glucose tolerance test (Supplementary Figure 2A), insulin tolerance test (data not shown), glucose transporter expression (GLUT1 and GLUT4 Supplementary Figure 2B and C) and the skeletal muscle metabolic phenotype (fed and fasted state Supplementary Figure 2D and E) were comparable between two genotypes. We have however amended our manuscript to include this comment (line 130-134).

2. Some of the figures are hard to interpret. The inclusion of color helps some, but they still run together (e.g. Fig. 1G). It would help to spread the individual metabolites to include white space (there is plenty available to the right as the figure is currently laid out) and/or add dotted lines between them. Figure 2D is slightly better in this regard.

We agree with the Reviewer and have amended our Figure 1G taking onboard their criticism.

Reviewer #3 (Remarks to the Author):

This is a nice study showing that acute elevation of intracellular [Na]_i in cardiac myocytes causes a metabolic shift (independent of increased work) quite similar to some known to occur in heart failure (substrate shift from fatty acids toward glucose utilization). The aspect of the results that is especially remarkable is that even the acute Na/K-ATPase (NKA) inhibition causes this shift in minutes, and that the effects are similar to those seen in a chronic phospholemman (PLM) knockin mouse that inhibits NKA and raises [Na]_i and also a pressure-overload model of cardiac hypertrophy. These metabolic changes have usually been assumed to require transcriptional regulation and altered gene expression, but this study suggests that they are acutely caused by elevated [Na]_i (and presumably reduced [Ca]_{mito}) before major gene transcription effects are expected. I do have a number of suggestions that could strengthen the paper and make the conclusion more convincing.

We thank the Reviewer for the thorough critique of our work and thank them for finding aspects of our study “remarkable”. We too were surprised that these effects can be acutely manipulated. We expected substrate preference in particular to be longer-term transcriptionally dependent. It genuinely surprised us too! We don’t doubt that more long-term transcriptional changes also occur and we are sure that our present observations simply add to a much more complex scenario in heart failure.

Main points:

1. The most exciting aspect of this study is the possibility that these metabolic shifts might occur acutely by direct consequences of [Na]_i and possibly mitochondrial [Ca] via acute regulation of the existing protein machinery, before substantial changes in gene regulation and consequent protein expression could have occurred. To strengthen this conclusion, it would be valuable to show that:

a) myocyte protein expression (with respect to a panel of key proteins) does not change appreciably in the 30 min ouabain-blebbistatin case or CGP exposure vs. control,

We agree with the Reviewer.

We have examined the expression of isocitrate dehydrogenase (TCA cycle Ca-sensitive rate determining step regulator enzyme) and pyruvate dehydrogenase (PDH) in our acute (Oubain+Blebbistatin) and chronic sodium (PLM^{35A}) elevation groups. We found that there is no change in expression of either of the enzymes in agreement with no change in mRNA levels in a series of metabolic regulators (Supplementary Figure 3). We have now included this finding in the main manuscript (lines 194-195) and in the supplementary material (red font) (Online Supplement lines 48-50, 392-396, Supplementary Figure 4).

Figure 2: IDH and PDH expression in the acute and chronic sodium elevation (New Supplementary Figure 4)

b) another acute mechanism that should lower mitochondrial [Ca] has similar acute effects on metabolism (e.g. Ru360) as Ouabain-Blebbsi,

We have taken this comment on board and examined the literature for the suitable Ru360 perfusion protocol to add to our study. However, we have found that in the isolated perfused heart, Ru360 only partially inhibits the mitochondrial Ca uniporter thus may not lower Ca lower Ca effectively (Jesus Garcia-Rivas et al 2005 FEBS J, 272: 3477-3488).

c) [Ca]mito is actually reduced by acute [Na]i elevation and reversed by CGP.

Yes – this is the hypothesised basis of these effects and those described previously by O'Rourke and colleagues (Lie et al 2014 Circ Res, Liu and O' Rourke Circ Res 2008, Liu et al JMCC 2010, Kohlhaas et al. Circ 2010).

d) the computational model explains this mechanistically by simple [Ca]mito and/or [Na]i changes. That might have been the aim, but as written it totally misses the mark (see below).

We apologise and accept that this section was not written clearly. We have now rewritten this (both in Methods, and Results) in order to clarify the use of Cardionet modelling (also see comment 4 below)

2. Ln 179-182. In Fig 1 CGP alone seemed to cause a slight downward shift in the FA/Glucose utilisation, compared to control. While the individual effects (FA or Glu) were not significantly different, this ratio seems to go in the opposite direction to that expected from the main thesis. And in line 186-7 there do seem to be some CGP effects on metabolites vs. WT.

We thank the reviewer for this observation, however statistical analysis of the fuel utilization and metabolite levels do not show any significant metabolic changes caused by CGP37157 in WT or control hearts thus we cannot speculate on trends. Furthermore, hearts were perfused in the presence of other carbohydrates and amino acid (pyruvate, lactate, glutamate) thus we cannot simply analyse the ratio of glucose to palmitate utilization as the presence of other unlabelled substrates as well as endogenous pool of glycogen and triglycerides will complicate this analysis.

3. It seems important that CGP effects in Fig 2 (like ouabain effects later) were manifest in 30 min of exposure. This relates to point #1 above, and may be worth highlighting in that context (acute regulation vs. transcription).

We agree with the Reviewer. We have added this comment to our manuscript (line 150).

4. The CardioNet computer model and its use are very poorly described and hence difficult to evaluate. The reference in the main manuscript (24) does not even mention CardioNet (although

O'Rourke & Maack do mention the Cortassa et al. model), nor are the different references in the Supplement (10,11) primary papers that developed CardioNet. There is no acknowledgement as to the strengths or weaknesses of the model, or its precision with respect to mouse ventricular myocytes. It also sounds like various things in the model were adjusted, but the reasoning or justification is not clear (and that should be distinguished clearly from the use of the model). If you manipulate the model to fit your data, you can no longer use it as a "logical" test of plausibility.

We thank the reviewer for making us aware of these discrepancies. We revised the manuscript to provide the reader more information on constrained-based and genome-scale metabolic networks (see main text and Supplemental Materials). CardioNet has been extensively validated and has been successfully applied in three other studies to assess metabolic changes in cardiomyocyte metabolism, which are all cited in the main text of the manuscript. The primary article describing the reconstruction and validation of CardioNet has been cited on pages 9 and 10 of the main text, and pages 7 to 8 in the supplementary materials. Please see the following published work:

1. Oncometabolite d-2-hydroxyglutarate impairs α -ketoglutarate dehydrogenase and contractile function in rodent heart. Karlstaedt A, Zhang X, Vitrac H, Harmancey R, Vasquez H, Wang JH, Goodell MA, Taegtmeier H. Proc Natl Acad Sci U S A. 2016 Sep 13;113(37):10436-41. doi: 10.1073/pnas.1601650113. Epub 2016 Aug 31. PMID: 27582470
2. Cardiac dysfunction and peri-weaning mortality in malonyl-coenzyme A decarboxylase (MCD) knockout mice as a consequence of restricting substrate plasticity. Aksentijević D, McAndrew DJ, Karlstädt A, Zervou S, Sebag-Montefiore L, Cross R, Douglas G, Regitz-Zagrosek V, Lopaschuk GD, Neubauer S, Lygate CA. J Mol Cell Cardiol. 2014; 75:76-87. doi: 10.1016/j.yjmcc.2014.07.008. Epub 2014 Jul 24. PMID: 25066696
3. CardioNet: a human metabolic network suited for the study of cardiomyocyte metabolism. Karlstädt A, Fliegner D, Kararigas G, Ruderisch HS, Regitz-Zagrosek V, Holzhütter HG. BMC Syst Biol. 2012;6:114. doi: 10.1186/1752-0509-6-114.

Additionally, we validated *CardioNet* simulations by comparing estimated flux rates with experimentally measured rates. This comparison is described in the main text of the manuscript lines 237-263.

Genome-scale metabolic networks like *CardioNet* allow prediction of flux distributions using constrained-based modeling once an equilibrium or steady state has been reached. The reviewer asserts that "various things in the model were adjusted, but the reasoning or justification is not clear (and that should be distinguished clearly from the use of the model)." We would like to respectfully point out that this statement is not correct. We did not "adjust" or "change" the model. We state on p. 7 and 8 in the Supplemental Materials that "simulations were run with boundary conditions reflecting the metabolite composition of the perfusion buffer and experimentally measured uptake and release rates of substrates". The model *CardioNet* provides a tabulation of metabolic reactions that allow mathematical simulations using f.exp. flux balance analysis. This numerical representation of reactions or stoichiometric matrix imposes constraints on the flow of metabolites through the network. Flux balance analysis also allows to impose bounds on every reaction. These upper or lower bounds define the maximum and minimum allowable fluxes of the reactions that are consistent with the experimentally measured inputs. Irreversible reactions have a lower bound that is equal to zero. As described in the Supplemental Materials, bounds are not imposed on fluxes unless the reactions are "experimentally measured uptake [or] release rates of substrates".

We have now made every effort to improve the representation of CardioNet including previous work throughout the manuscript. We describe the model and algorithms for flux balance analysis in the supplementary methods. However, for a comprehensive assessment of the model capabilities we would like to refer to Karlstaedt et al. BMC Syst Biol 2012 and our previous work. We further revised the text to provide readers with more information on constraint-based modeling and added additional references to provide clarity (red font, pages 9-11).

It seems to me that the main value of the model in this study would be to test whether the simple change in [Na]_i or [Ca]_{Mito} by itself (e.g. by known effects on dehydrogenases) would be expected based on the existent model to cause the metabolic changes observed. The only things changed should be either [Na]_i, [Ca]_{Mito} or if CardioNet does not have good mitochondrial Na-Ca flux descriptions you could change activity of the specific Ca-dependent processes. Maybe that is what you are doing (ln 258, 70, 80 and 90%) but even that is unclear. On line 262-3 a 0.7 fold decrease in flux is stated; is this a reduction to 70% of V_{max} (30% remaining) or a reduction to 70% V_{max}. Also flux should be an output that depends not only on V_{max}, but also levels of substrates and products. This section really needs a major revision if it is to be useful. Ln 323-326 should be more explicit about the modeling as a plausibility test of the Na-Ca hypothesis you propose.

CardioNet is a metabolic network of mammalian cardiac metabolism, thus does not allow to simulate sodium or calcium dynamics directly. As the reviewer correctly points out, we simulated how increased sodium levels impair Ca-dependent reactions with the goal to identify which metabolic processes are involved in the adaptation. Flux simulations are dependent on the model matrix (stoichiometric matrix), which includes substrates and products for each reaction, boundary conditions for specific substrate rates, and the objective function.

We would like to use this opportunity to emphasize that V_{max} is not equivalent with flux. In general, it cannot be concluded that a predicted flux rate change of 0.7 is equivalent to 70% V_{max} reduction because the V_{max} of a specific enzyme may increase exponentially rather than linear. A good example here are enzymes with Hill kinetics such as phosphofructokinase. The enzymatic capacities for flux (or V_{max}) do not match a pathway flux, but instead can greatly exceed pathway flux rates. In other words, a flux rate can change while the V_{max} or capacity of the corresponding enzyme is not affected while an inhibitor (e.g. product) increased. A good example is hexokinase where the V_{max} may differ by 100-fold between resting and exercise conditions, but the actual flux rate change may be 80,000-fold. We have now made every effort to clarify how simulations were conducted and which conclusions can be drawn from our analysis.

5. The CGP37157 concentration and its expected (or measured) level of NCLX inhibition should be stated. It can only be partial inhibition, because as we now know complete NCLX knockout is lethal (PMID: 28445457). See also point 1c above.

The CGP concentration used in the perfusate (1 μmol/l) is given in our Methods section as well as figure legends. However, how much of this actually reaches the mitochondria is clearly unknown and how that translates into NCX inhibition is equally unknown. In isolated mitochondria, the IC₅₀ of CGP is 0.40 μM. (Cox et al 1993). Given the lethality of the KO described by Luongo et al, we agree that complete inhibition of NCLX may be catastrophically lethal. However, in our studies it seems that partial inhibition or even transient complete inhibition is survivable in the short term. Unlike measuring surrogate markers, as far as we are aware there is no direct way to test the extent of NCLX inhibition in an intact heart. It is clear, however, that even if the NCLX inhibition is

partial, it is sufficient to acutely prevent the metabolic switch. We have now discussed this issue in the revised manuscript (lines 378-384).

Minor:

Pg 1. Author affiliations for ALL authors should be indicated (by superscript).

Manuscript adjusted (title page).

Pg 2. Last line in Abstract should be "...be a new approach..."

Manuscript adjusted: abstract had to be reduced to 150 words.

Pg 8: It may be worth indicating that the rise of [Na]_i in heart failure may be due in part to elevated late Na current (many refs), diastolic Na influx via Na channels (Despa et al. 2002) and/or reduced NKA function (author's work?).

We thank the reviewer for this comment which has been added to our introduction (lines 90-92).

Reviewers' comments:

Reviewer #1 (Remarks to the Author):

The authors have been responsive. Just a couple of minor points. First, lines 317 and 320 refer to concentrations of metabolites (using the terms “depletion” and “levels”) and remark on “utilization” and “remodeling”. I think what they are concluding is that fluxes can be inferred from metabolite concentrations, which is incorrect. For example, a low concentration of lactate does not provide any information about anaplerotic flux. Also, line the sentence beginning on line 346 is a little confusing. The reference to a “substrate switch” away from fatty acid oxidation implies that another substrate is being used to supply acetyl-CoA, which could be ketones, pyruvate, some amino acids, etc. But the reference to “anaplerotic fuels” implies metabolism through a non-oxidative pathway. This could be clarified.

Reviewer #2 (Remarks to the Author):

Initially I found the manuscript of merit. With the responses to my minor comments and the extensive responses to the comments of the other reviewers I believe this has been further strengthened and have no significant concerns of note.

Reviewer #3 (Remarks to the Author):

The authors have been responsive to the prior critiques and the manuscript is substantially strengthened. The additional experimental work and controls make the overall conclusions even more convincing. However, the Computational Modeling section remains poorly developed. They added some more words, that make it somewhat more clear what they have done. However, it still lacks clarity as to what was done and what it means, even to this reviewer who has extensive experience in cardiac computational modeling. For the general readership it is likely to be totally opaque and likely ignored. I have some remaining general comments, but will especially focus constructively on how the modeling component could be strengthened.

Major

Line 146-50: It would be expected that CGP37517 should raise mitochondrial Ca, even in control myocytes, which ought to lead to an effect opposite to that seen in Fig 1A with $[Na]_i$ elevation induced $[Ca]_{mito}$ reduction. A question related to Fig 1A and 2B is whether the apparent inability to detect a significant difference for CGP treatment in PLM-S3A or Banded was due to sample size being small (especially in 2B). No comparison bars are shown.

Line 150-52 : Fig S6 legend says 30 min while figures labels say 50 min. Please correct. The A, B and C panels are also not specified as to how they differ. All the labeling is the same (although font sizes are far too small). If A-C are in different myocytes or conditions, that should be specified.

Line 208-10: It seems like it is worth noting the huge increases in lactate & Isocitrate and the big drop-off after alphaKG. Moreover, it would be worth noting how these effects would be consistent with reduced Ca-dependent activation of the critical dehydrogenases at these points.

Page 9-11 (Modelling): The first reference here to CardioNet (ln 225) still refers to a wrong reference (a review by S. Neubauer that does not mention this model). I guess you mean to cite ref 20. I'm not sure whether refs 18-20 use CardioNet although that is implied (ln 226).

This whole first paragraph could be shortened and more focused on how you are using the model. In this version I think I understand better the two things you are doing (I & ii in ln 233-36) but it is still not well communicated. I suggest that you start the second paragraph of this section with:

“First, we used CardioNet to predict the metabolic impact of a simple 30% reduction of Ca-dependent activation of three different dehydrogenases (PDH, isocitrate dehydrogenase and aKG dehydrogenase). This is meant to simulate the effects of elevated $[Na]_i$ to reduce $[Ca]_{Mito}$ and thus these dehydrogenases. When we allow the model to reach steady state with no other parameter changes, the predicted flux rate changes of other enzymes are shown in Fig S7. An overall decrease in Krebs's cycle flux occurred (mostly at 70-80% of control), except for citrate synthase which was greatly elevated. Both glucose and fatty acid pathway fluxes were also reduced, but NADH and FADH₂-dependent ATP synthesis was not compromised.”

You can clarify in the supplement or Methods that you are changing the dynamic range, rather than the Ca-dependence of the dehydrogenase activity (if indeed that is a correct inference of your statement). You should be specific as to whether the values shown are “flux rates” or “flux rate constants”. At the end of this suggested passage, I think you need to be clearer about how the model actually manages to maintain OXPHOS in the presence of reduced flux via Glycolysis, TCA cycle, KB and BCAA. Why does SOD shut off? Things here do not quite add up (what fuel is driving the normal ATP generation if all fluxes are reduced?). I also think you should show how the metabolic profile measured in Fig 4B is matched by the model results (a second panel in Fig S7). This would allow you to assess whether simply lowering these Ca-dependent dehydrogenases suffices to

mimic the Ouab+Blebb, and that's how this paragraph should end. Maybe Fig S7 should be a full Fig 5?

The next paragraph should start something like this:

"A second modeling approach used the actual metabolite measurements in high $[Na]_i$ cases to iteratively constrain the flux rates and determine those that could explain that metabolite fingerprint. The CardioNet model converged on the properties illustrated in Fig 5..."

Maybe this is not what you did, but I worry that the acute high $[Na]_i$ case in Fig 5B looks to be diametrically opposed to your data in Fig S7, and even the Ca-dependent dehydrogenases are highly increased. Indeed, the whole TCA system is ramped up ~2-fold, as is carbohydrate flux and ATP production, while FA flux is reduced. While this substrate shift matches the data in Fig 1 and 4, Fig S7 doesn't match that at all, but may be consistent with the build-ups of substrate behind the dehydrogenases (data that I suggest you show). Moreover in Fig 5B there is increased flux through all of the Ca-dependent dehydrogenases? In the end, this section still ends up being confusing both in what it means and how it connects to the data. Making this section clearer may also help the discussion. There are opportunities for actual validation (predicted metabolites matching model output in Fig S7) but those are not developed.

Fig 5 A-B are also too small to read the labels and numbers in the printed figure.

Minor

Line 43 should read "Intracellular Na elevation in heart..."

Line 126 do you want to see N_{ai} rather than $[Na]_i$?

Line 155: perhaps "...chronic N_{ai} elevation (PLM-S3A vs. PLM-WT) leads to..."

Line 175-77: Except for Asp and Ala??

Line 316-19: This CardioNet sentence needs to match the data shown. How well does the $[Na]_i$ dependent change in $[Ca]_{Mito}$ and dehydrogenase function replicate the experimental results.

Line 335: Revise to "...myocardial N_{ai} elevation (and consequent $[Ca]_{Mito}$ decline) has the potential to reverse metabolic derangement..."

Line 348 "corroborated" should be "consistent with"

Line 354: All substrates are exogenous, and it is unclear what you mean here.

Reviewers' comments:

Reviewer 1

The authors have been responsive. Just a couple of minor points. First, lines 317 and 320 refer to concentrations of metabolites (using the terms “depletion” and “levels”) and remark on “utilization” and “remodeling”. I think what they are concluding is that fluxes can be inferred from metabolite concentrations, which is incorrect. For example, a low concentration of lactate does not provide any information about anaplerotic flux. Also, line the sentence beginning on line 346 is a little confusing. The reference to a “substrate switch” away from fatty acid oxidation implies that another substrate is being used to supply acetyl-CoA, which could be ketones, pyruvate, some amino acids, etc. But the reference to “anaplerotic fuels” implies metabolism through a non-oxidative pathway. This could be clarified.

We thank the Reviewer for these points and completely agree. We have now corrected the text (red font) in lines 284 (page 11). Specifically, we have removed any reference to flux estimates based solely on measured concentrations. We have also removed references here to anaplerotic ‘flux’ and listed the metabolites we experimentally assessed in terms of their concentrations (metabolomic profiling, 31P NMR spectroscopy), utilization (¹³C NMR spectroscopy substrate contribution analysis) and flux balance analysis (Cardionet modelling).

Reviewer #3 (Remarks to the Author):

The authors have been responsive to the prior critiques and the manuscript is substantially strengthened. The additional experimental work and controls make the overall conclusions even more convincing.

However, the Computational Modeling section remains poorly developed. They added some more words, that make it somewhat more clear what they have done. However, it still lacks clarity as to what was done and what it means, even to this reviewer who has extensive experience in cardiac computational modeling. For the general readership it is likely to be totally opaque and likely ignored. I have some remaining general comments, but will especially focus constructively on how the modeling component could be strengthened.

We apologise for not making this clearer and we have gone back and substantially changed the way this is described, presented and indeed analysed. We have taken the Reviewer’s criticisms and suggestions on board and have substantially altered our Cardionet modelling section and we are striving to make it understandable (all changes made are in red font). In order to try to improve the clarity, we have replotted all our modelling data (new figure 5, 6) as well as generated new supplementary figures 7, 8 and 9 which we hope demonstrate the model in a clearer and more straight-forward way.

Major

Line 146-50: It would be expected that CGP37517 should raise mitochondrial Ca, even in control myocytes, which ought to lead to an effect opposite to that seen in Fig 1A with [Na]_i elevation induced [Ca]_{mito} reduction.

This is a very interesting point. Indeed, the opposite effect would clearly be predicted if the relationship between NCLX activity and intra-mitochondrial Ca was linear. However, should this relationship be significantly non-linear then it is possible that lowering activity below basal conditions does not do the opposite to increasing activity under Na-loaded conditions. While there are some small non-significant effects of CGP on basal metabolism, we agree that these are clearly not 'opposite' to the effects when NCLX is activated. This non-linearity may reflect the non-linearity of the very powerful buffering effects of mitochondrial Ca²⁺PO₄ (Israel Sekler – personal communication). While there is a wealth of literature (including in the heart – see papers from Maack, O'Rourke etc) showing that CGP affects metabolism when NCLX is activated, there is surprisingly no evidence showing CGP elevates mito Ca basally. On the contrary, Jornot et al (J Cell Sci: 1999 (112), 1013-1022: i.e. Fig 5) show that CGP does not elevate basal mitochondrial matrix Ca (measured with mito-targeted aequorin) when applied to a resting cell but does affect matrix Ca when the system is perturbed. This issue is now briefly discussed in the revised manuscript (line 299-305, page 12).

A question related to Fig 1A and 2B is whether the apparent inability to detect a significant difference for CGP treatment in PLM-S3A or Banded was due to sample size being small (especially in 2B). No comparison bars are shown.

We agree. The scatter in these measurements is quite large and it is possible we are missing some real effects of CGP37157 on baseline metabolism. However, using ANOVA these changes are not statistically significant except Banded glucose vs Banded CGP37157 glucose % contribution (P<0.01) which is normalized with the treatment. We have added the comparison bars to our Figures 1A and 2B. We have also added a paragraph in the text (page 12) discussing why CGP has limited effect on metabolism at baseline and does not seem to do the opposite of Na elevation (see the previous point). We also point out the limitations imposed by sample size and the very real possibility of Type II errors.

Line 150-52 : Fig S6 legend says 30 min while figures labels say 50 min. Please correct. The A, B and C panels are also not specified as to how they differ. All the labeling is the same (although font sizes are far too small). If A-C are in different myocytes or conditions, that should be specified.

We apologize for not making the supplementary Figure 6 more clear. We have increased the font size and removed the labelling A-C as these functional parameters have been acquired under identical conditions. We have also added the non-significant labels (bars).

Line 208-10: It seems like it is worth noting the huge increases in lactate & Isocitrate and the big drop-off after alphaKG. Moreover, it would be worth noting how these effects would be consistent with reduced Ca-dependent activation of the critical dehydrogenases at these points.

We thank the Reviewer for this point- we have now amended the manuscript to address. Page 8, 203-207

Page 9-11 (Modelling): The first reference here to CardioNet (In 225) still refers to a wrong reference (a review by S. Neubauer that does not mention this model). I guess you mean to cite ref 20. I'm not sure whether refs 18-20 use CardioNet although that is implied (In 226).

We apologize for this Endnote referencing software lapse which we have now corrected as this section has been extensively rewritten. The references in question are relevant to Cardionet use in previous cardiac metabolism studies.

This whole first paragraph could be shortened and more focused on how you are using the model. In this version I think I understand better the two things you are doing (I & ii in In 233-36) but it is still not well communicated. I suggest that you start the second paragraph of this section with:

“First, we used CardioNet to predict the metabolic impact of a simple 30% reduction of Ca-dependent activation of three different dehydrogenases (PDH, isocitrate dehydrogenase and aKG dehydrogenase). This is meant to simulate the effects of elevated $[Na]_i$ to reduce $[Ca]_{Mito}$ and thus these dehydrogenases. When we allow the model to reach steady state with no other parameter changes, the predicted flux rate changes of other enzymes are shown in Fig S7. An overall decrease in Krebs's cycle flux occurred (mostly at 70-80% of control), except for citrate synthase which was greatly elevated. Both glucose and fatty acid pathway fluxes were also reduced, but NADH and FADH₂-dependent ATP synthesis was not compromised.”

We thank the reviewer for these points. In the revised manuscript we have now decided that the narrative can be significantly simplified by removing the 30% inhibition simulations entirely. These were our initial attempt to understand whether this could replicate the

features in our data. Clearly this simulation does not help in this regard. We feel that these simulations introduce confusion and therefore we now limit the modelling only to the model constrained by the actual data. We made extensive changes to include the Reviewer's suggested narrative. The metabolic network CardioNet does not include calcium (nor any other electrolyte). Therefore, we cannot directly model calcium dependencies, as the reviewer correctly pointed out. We have included a statement in the main text and the supplementary material to clarify this aspect to readers.

Supplementary material pages 7-9, manuscript pages 9-12.

You can clarify in the supplement or Methods that you are changing the dynamic range, rather than the Ca-dependence of the dehydrogenase activity (if indeed that is a correct inference of your statement). You should be specific as to whether the values shown are "flux rates" or "flux rate constants".

As indicated in the main text and supplemental methods, we calculated "flux rates" using flux balance analysis. We do not refer to rate constants. Flux balance analysis follows the law of mass action but does not include rate constants in the calculations. For example, for a reaction like: $S_1 + S_2 \leftrightarrow 2P$ the reaction rate is: $v = v_+ + v_- = k_+ S_1 * S_2 - k_- P^2$, where v is the net flux rate, v_+ the rate of the forward reaction, v_- the rate of the backward reaction, and k_+ and k_- are the kinetic or rate constants. As indicated in the Supplemental Materials, we do not include rate constants or rate equations in our calculations.

Supplemental Methods pp. 7-9.

At the end of this suggested passage, I think you need to be clearer about how the model actually manages to maintain OXPHOS in the presence of reduced flux via Glycolysis, TCA cycle, KB and BCAA. Why does SOD shut off? Things here do not quite add up (what fuel is driving the normal ATP generation if all fluxes are reduced?). I also think you should show how the metabolic profile measured in Fig 4B is matched by the model results (a second panel in Fig S7). This would allow you to assess whether simply lowering these Ca-dependent dehydrogenases suffices to mimic the Ouab+Blebb, and that's how this paragraph should end. Maybe Fig S7 should be a full Fig 5?

The next paragraph should start something like this:

"A second modeling approach used the actual metabolite measurements in high $[Na]_i$ cases to iteratively constrain the flux rates and determine those that could explain that metabolite fingerprint. The CardioNet model converged on the properties illustrated in Fig 5..."

Maybe this is not what you did, but I worry that the acute high $[Na]_i$ case in Fig 5B looks to be diametrically opposed to your data in Fig S7, and even the Ca-dependent dehydrogenases are highly increased. Indeed, the whole TCA system is ramped up ~2-fold, as is carbohydrate flux and ATP production, while FA flux is reduced. While this substrate shift matches the data in Fig 1 and 4, Fig S7 doesn't match that at all, but may be consistent with the build-ups of substrate behind the dehydrogenases (data that I suggest

you show). Moreover in Fig 5B there is increased flux through all of the Ca-dependent dehydrogenases?

We thank the reviewer for making us aware of these discrepancies and apologize for any misleading statements. We have now thoroughly revised this section in the main text of the manuscript and the method section.

We hypothesized that $[Na]_i$ elevation causes inhibition of Ca-dependent dehydrogenases based on previously discussed mechanisms (see Literature Cox and Matlieb, Maack and O'Rourke). However, these molecular mechanisms have never been tested under physiological conditions. We conducted modeling based on the experimental metabolic profiles during acute and chronic $[Na]_i$ elevation.

The set of simulations constrained to experimental data are depicted in Figure 5 and 6 and and Figure S7-9. The experimental data used to constrain the model comprised of the metabolomic profiles, ^{13}C substrate utilization, ^{31}P NMRS energetics in acute (blebbistatin+ouabain) and chronic Na elevation (PLM^{3SA}). The modeling shows that the impact of impaired Na_i homeostasis on mitochondrial ATP production is mechanistically more complex than previous studies suggested using isolated cells and organelles and that there is no evidence of an energetic deficit. In fact, the simulations showed that ATP provision for contractile work is maintained due to compensatory increase of amino acid and carbohydrate utilisation (pyruvate, lactate) to replenish Krebs cycle intermediates. In agreement with our experimental data, modelling shows that the contribution of fatty acid β -oxidation to ATP provision decreased in response to acute and chronic Na elevation.

We revised the manuscript as follows:

1. We replaced Figure 5 with two new figures a more detailed depiction of the flux distributions in response to elevated sodium. (acute and chronic)
2. To provide more clarity, we removed supplementary figure 7. Given that we didn't run experiments that study the impact of dehydrogenase inhibition on cardiac metabolism and/or Ca flux thus we cannot corroborate our modelling.

In the end, this section still ends up being confusing both in what it means and how it connects to the data. Making this section clearer may also help the discussion. There are opportunities for actual validation (predicted metabolites matching model output in Fig S7) but those are not developed.

We thank the reviewer for raising these important concerns. We have now included a comparison between the experimentally determined metabolic profile (acute sodium elevation, Figure 4B) to the profile generated by the simulations of acute Na elevation. As indicated in this new Figure S9, simulation results and experimental data strongly correlate with each other. The model correctly predicted metabolic changes based on the experimental profile during acute and chronic $[Na]_i$ elevation.

Fig 5 A-B are also too small to read the labels and numbers in the printed figure.

We revised Figure 5 and for clarity have divided into two figures 5 and 6.

Minor

We thank the Reviewer for these points. All suggestions taken on board and changes made to the manuscript (changes in red font).

Line 43 should read “Intracellular Na elevation in heart...” changed line 41
Line 126 do you want to see Na_i rather than $[\text{Na}]_i$? –changed line 126

Line 155: perhaps “...chronic Na_i elevation (PLM-S3A vs. PLM-WT) leads to...” line 151, pg6

Line 175-77: Except for Asp and Ala??

Changed – lines 170,171

Line 316-19: This CardioNet sentence needs to match the data shown. How well does the $[\text{Na}]_i$ dependent change in $[\text{Ca}]_{\text{Mito}}$ and dehydrogenase function replicate the experimental results.

New paragraph has been added to pg 11-12 lines 284-294

Line 335: Revise to “...myocardial Na_i elevation (and consequent $[\text{Ca}]_{\text{Mito}}$ decline) has the potential to reverse metabolic derangement...”

Inserted Pg13 lines 321-322

Line 348 “corroborated” should be “consistent with”

Changed Pg 13, 334

Line 354: All substrates are exogenous, and it is unclear what you mean here. Changed to “metabolic substrate”.

Changed 340, pg 14

Reviewers' comments:

Reviewer #3 (Remarks to the Author):

This "R2" revised manuscript, in which I thought the experimental work was excellent before, has been further improved, such that the experimental tests of their implicit core hypothesis in Figs 1-4 (and S1-S6) are very nicely and clearly described. Outstanding! Their implicit hypothesis is that elevated $[Na]_i$ causes metabolic changes that are independent of work, but are dependent upon the effect of high $[Na]_i$ to reduce mitochondrial $[Ca]$. The fact that inhibition of the mitochondrial Na/Ca exchanger (NCLX) to promote higher mitochondrial $[Ca]$ nicely restores the metabolic profile for both acute and chronic $[Na]_i$ elevation makes a strong test case for this innovative hypothesis. Furthermore, since mitochondrial dehydrogenases are well-known to be regulated by levels of $[Ca]$, and high $[Na]_i$ causes substrate accumulation upstream of both PDH and α -Ketoglutarate dehydrogenase, that conclusion makes excellent mechanistic sense. Although the findings here do not elucidate the detailed mechanism for the switch from fatty acid to glucose utilization, the experimental metabolic profiles might help future hypothesis building. This core of the paper is very strong.

I was (and remain) more critical of the CardioNet modelling, despite the authors' strong efforts and major changes to improve the clarity of this section (Fig 5-6 and S7-S9) in the manuscript and Online Methods. That helps, but I still find this part highly disappointing in several ways. There were 2 modelling parts before. The most relevant part was where they reduced the function of the 3 well studied Ca-dependent dehydrogenases (PDH, α KGDH & Isocitrate DH) to test whether the $[Na]_i$ -dependent $[Ca]_m$ changes might explain the metabolic profile measured. That was, indeed, right "on-target" to test the central hypothesis of the study. I had focused my suggestion to encourage them to strengthen this hypothesis-testing aspect. Even if such decreases in activation state of these 3 DHs did not produce the full tableau of metabolic profiles, it might point toward what else might have to occur to get there – i.e. helping develop new hypotheses for future testing (e.g. an unrealized $[Ca]_m$ - or α KG- sensitive cause of shift from FA to Glucose utilization).

In the revised manuscript they have deleted that aim-appropriate use entirely. Instead they use CardioNet and make changes in metabolite concentrations to their measured values and see which enzymes are predicted to change when the model runs to optimize steady state ATP production (an arbitrary goal). While that could be useful, such a solution is not unique, does not test a mechanistically relevant hypothesis, and in this case is fundamentally at odds with the central hypothesis regarding how $[Na]_i$ may alter Ca-dependent dehydrogenases. They use this weak reasoning to undercut their central conclusion, in what I think overstates the power of the model (or its use here). The first results in Fig 5 are also still not clearly described and raise many questions. What do the z-scores represent? Does a positive z score imply the enzyme is producing higher flux than some control or baseline condition? Why does the baseline already show a wide array of heat-

map changes? Does this simply indicate how different their baseline metabolite values are when compared to those that are embedded in the canned model (i.e. is this how badly CardioNet predicts their baseline data?)? Do the Na elevations (acute & chronic) imply changes from baseline or changes from the canned model fluxes? As it stands, the modelling takes some very nice mechanistically focused experimental data and dissipates that into a diffuse manuscript with very fuzzy conclusions.

A point in the authors response (and the prior Figure S7) also gives me concern as to the level of mechanistic insight that can be gained by the model. In their Responses they state that they don't consider rate constants and are using "flux rates" and that "Flux balance analysis follows the law of mass action but does not include rate constants in the calculations." Then they show a normal mass-action equation that includes rate constants. If CardioNet does not use rate constants (which include substrate affinities and activity), how can they change the activation state of these dehydrogenases? That explains why in their prior Fig S7 they change the FLUXES by 30% rather than the rate constants of enzyme. The forward reaction rate must be proportional to both the substrate concentration and activity (e.g. v/V_{max}). If the activity is reduced by 30%, this would tend to cause build-up of substrate behind this step, but that increased substrate concentration would increase Flux through that enzyme until a new steady state is achieved, and that is influenced by all of the other reactions in the network. If the model does not have this power, then it is not really appropriate for testing the feasibility of the central hypothesis here.

Response to Reviewer #3

We thank the reviewer for their very supportive comments regarding the experimental studies and our efforts to improve the clarity of our manuscript. In addition to answering the comments (see below) we have also now included a section in the revised manuscript entitled "*Limitations of the current study*". (Main Manuscript: lines 376-402). This provides us with the opportunity to discuss issues raised here and to identify future areas for hypothesis-testing (see discussion below).

Reviewer's Comment:

This "R2" revised manuscript, in which I thought the experimental work was excellent before, has been further improved, such that the experimental tests of their implicit core hypothesis in Figs 1-4 (and S1-S6) are very nicely and clearly described. Outstanding! Their implicit hypothesis is that elevated [Na]_i causes metabolic changes that are independent of work, but are dependent upon the effect of high [Na]_i to reduce mitochondrial [Ca]. The fact that inhibition of the mitochondrial Na/Ca exchanger (NCLX) to promote higher mitochondrial [Ca] nicely restores the metabolic profile for both acute and chronic [Na]_i elevation makes a strong test case for this innovative hypothesis. Furthermore, since mitochondrial dehydrogenases are well-known to be regulated by levels of [Ca], and high [Na]_i causes substrate accumulation upstream of both PDH and α-Ketoglutarate dehydrogenase, that conclusion makes excellent mechanistic sense. Although the findings here do not elucidate the detailed mechanism for the switch from fatty acid to glucose utilization, the experimental metabolic profiles might help future hypothesis building. This core of the paper is very strong.

Authors' Response:

Thank you for these supportive comments.

Reviewer's Comment:

I was (and remain) more critical of the CardioNet modelling, despite the authors' strong efforts and major changes to improve the clarity of this section (Fig 5-6 and S7-S9) in the manuscript and Online Methods. That helps, but I still find this part highly disappointing in several ways. There were 2 modelling parts before. The most relevant part was where they reduced the function of the 3 well studied Ca-dependent dehydrogenases (PDH, αKGDH & Isocitrate DH) to test whether the [Na]_i-dependent [Ca]_m changes might explain the metabolic profile measured. That was, indeed, right "on-target" to test the central hypothesis of the study. I had focused my suggestion to encourage them to strengthen this hypothesis-testing aspect. Even if such decreases in activation state of these 3 DHs did not produce the full tableau of metabolic profiles, it might point toward what else might have to occur to get there – i.e. helping develop new hypotheses for future testing (e.g. an unrealized [Ca]_m- or αKG-sensitive cause of shift from FA to Glucose utilization). In the revised manuscript they have deleted that aim-appropriate use entirely.

Authors' Response:

This point can be summarised by the question:

1. Why did we remove, rather than improve, the "*on-target*" and "*aim-appropriate*" modelling simulating the metabolic consequences of reducing the 3 known Ca-sensitive dehydrogenases?

At the outset, one of the initial reasons for undertaking the modelling was to identify the possible role of the known Ca-sensitive dehydrogenases. Unfortunately, it became clear that limiting the modelling to adjusting only the fluxes of these 3 enzymes was naïve since this caused an overall reduction in Krebs cycle flux, a reduction in ATP production and a net decrease in fluxes entering the Krebs cycle which is at odds with what we observed experimentally - the reality is clearly much more complex. There may be many, as yet unknown, Ca-sensitive enzymes that need to be identified and their Ca-dependencies included in the model. We agree that ideally the modelling and the real measurements should be used iteratively to refine the model. However, this will require a more reductionist approach with the direct identification and measurement of Ca-dependent enzyme activities rather than complex in situ measurements of a host of interacting substrates, intermediates and end-products as well as any possible allosteric interactions. Although this is the strength of complex modelling, we would argue that this is beyond the scope of the present paper and complicates the relatively simple conclusions that the modelling allows us to reach none-the-less (see Point 4 below). We remain keen to take this forward (perhaps with refinements to CardioNet or using other available models) in future studies.

Despite its limitations we believe that CardioNet modelling remains very useful for giving additional insights into our experimental data. Notably through hierarchical clustering and PCA analysis it defines an unbiased metabolic 'fingerprint' that is common to both acute and chronic Na elevation and identifies shifts in substrate metabolism away from fatty acid oxidation towards anaplerosis. The flux balance analysis also allows us to identify which reactions are involved in metabolic adaptations to Na elevation. The heat maps in particular, we would argue, are a useful way to give an easy-to-visualise overview of these common pathways - making these conclusions more accessible to a general audience (see Point 3 below).

Reviewer's Comment:

Instead they use CardioNet and make changes in metabolite concentrations to their measured values and see which enzymes are predicted to change when the model runs to optimize steady state ATP production (an arbitrary goal). While that could be useful, such a solution is not unique, does not test a mechanistically relevant hypothesis, and in this case is fundamentally at odds with the central hypothesis regarding how $[Na]_i$ may alter Ca-dependent dehydrogenases. They use this weak reasoning to undercut their central conclusion, in what I think overstates the power of the model (or its use here).

Authors' Response:

This point can be summarised by the question:

2. Why did we optimize ATP production in CardioNet as an objective function for the flux balance analysis?

The decision to optimize steady-state ATP production was not "*arbitrary*". Our data show that while Na elevation re-programs metabolism, it does so while not compromising energetics – as demonstrated by our ^{31}P -NMR measurements showing maintained ATP, PCr, PCr/ATP ratios and pH_i . ATP and PCr concentrations are clearly a product of both production and

consumption, however, our wet-biology experimental design tries to keep consumption as constant as possible (by the titration of contractility with blebbistatin) and thereby keeping steady-state ATP production and consumption constant. However, we acknowledge that the activity of other ATPases, other than myosin ATPase, may complicate this picture: not least the quantitatively important energy consumption by SERCA. This is now discussed in the revised manuscript in the new section '*Limitations of the present study*' (main manuscript: lines 387-395).

We have also added discussion of why we chose to constrain the model in this way in the revised manuscript (see Supplement lines 313-319).

With regard to comments on the utility of the modelling, please see response below.

Reviewer's Comment:

The first results in Fig 5 are also still not clearly described and raise many questions. What do the z-scores represent? Does a positive z score imply the enzyme is producing higher flux than some control or baseline condition? Why does the baseline already show a wide array of heat-map changes? Does this simply indicate how different their baseline metabolite values are when compared to those that are embedded in the canned model (i.e. is this how badly CardioNet predicts their baseline data?)? Do the Na elevations (acute & chronic) imply changes from baseline or changes from the canned model fluxes? As it stands, the modelling takes some very nice mechanistically focused experimental data and dissipates that into a diffuse manuscript with very fuzzy conclusions.

Authors' Response:

This point can be summarised by the question:

3. What is the z-score, how are the heat maps derived and why are they useful?

We agree the z-scores were not well described. Familiarity made it seem clear to us while it was not to the reviewer. We apologize. Z-scores are calculated by averaging the absolute values of all the individual data points in each row of the heat map; including all control, acute and chronic Na elevation values. This gives us an average of all data points for that row which is equivalent to mean centering the data prior to performing the PCA analysis. The standard deviation for the entire row was then calculated. The average value in each individual 'cell' is then expressed in terms of the number of standard deviations it is from the mean for all the data points in that row. A z-score of 1 means that the measurement is 1 standard deviation away from the mean. The use of z-scores in this way is common practice in metabolomics and other omics data as absolute values can differ by orders of magnitude.

The wide array of changes in the baseline is not indicative of a large baseline variability but rather corresponds to reactions where the baseline is significantly higher or lower than the mean centred average (ie dark blue or bright yellow). These enzymes are the ones in which there are the biggest changes relative to the mean across all groups.

This data representation improves the readability of heat maps. Changes at baseline highlight the differences with respect to the treatments that are predicted by CardioNet. The model is not "*canned*" nor is it badly predicting baseline data. It is simply a mathematical centering of the data such that the mean falls between the control and treated groups. In fact, the model agrees well with our experimental data. We provide a comparison between predicted and experimentally determined metabolite fraction changes in Supplementary Figure 9 (see p. 20

Supplemental Materials). The Pearson's coefficient is 0.85. We would argue that the real take-home message of this is that it shows that Na elevation induces a metabolic fingerprint that is common irrespective of whether the Na elevation is acute or chronic. This is very easy to see in the heat-maps and the PCA plot in the Supplemental Data and which should make the take-home message clear to readers who are less familiar with the complexities of modeling, or with the complex patterns in multiple metabolic variables.

The calculation of z-scores in this way (as opposed to fold-changes with respect to a baseline) is a widely accepted and appropriate way of presenting metabolomic heat maps (and other omics data). We apologise for not making this clear in the previous manuscript. We have added clarification of how the z-scores were calculated (Supplement: lines 365-375). We have also amended the legend to Figure 5 to include a succinct explanation of how the heat map was derived (Main manuscript 726-738).

Reviewer's Comment:

A point in the authors response (and the prior Figure S7) also gives me concern as to the level of mechanistic insight that can be gained by the model. In their Responses they state that they don't consider rate constants and are using "flux rates" and that "Flux balance analysis follows the law of mass action but does not include rate constants in the calculations." Then they show a normal mass-action equation that includes rate constants. If CardioNet does not use rate constants (which include substrate affinities and activity), how can they change the activation state of these dehydrogenases? That explains why in their prior Fig S7 they change the FLUXES by 30% rather than the rate constants of enzyme. The forward reaction rate must be proportional to both the substrate concentration and activity (e.g. v/V_{max}). If the activity is reduced by 30%, this would tend to cause build-up of substrate behind this step, but that increased substrate concentration would increase Flux through that enzyme until a new steady state is achieved, and that is influenced by all of the other reactions in the network. If the model does not have this power, then it is not really appropriate for testing the feasibility of the central hypothesis here.

Authors' Response:

In addition to the specific point about rate constants, this point can be largely summarised by the question posed in the first line of the reviewers comment. That is:

4. Does CardioNet, as used, provide useful "mechanistic insights" and does it further our understanding?

Rate constants: In these previous simulations referred to (not present in the current manuscript) the reviewer is absolutely correct, we did not change the activity of the enzyme but rather the flux which, as stated, is the product of the rate constant and the concentration. Flux balance analysis does not include rate constants per se and therefore none of the equations presented include these. We are uncertain which equation the Reviewer is referring to when they state "*..they show a normal mass-action equation that includes rate constants*"?

With regard to the relationship between substrate concentration and flux this issue is complicated by the network of reactions that impinge both on the concentration of a substrate and the fate of the product – each reaction cannot be considered in isolation and is affected by interactions between enzymes and enzyme pathways. Together these factors can lead to

various combinations of enzyme-activity modulations but cause the same flux change. Other studies have addressed this important question to what extent metabolic fluxes are regulated by enzyme capacity (V_{max}) and to what extent by metabolic regulation (ie higher level global interaction and regulation) (see Rossell S et al. PNAS 2007;103:2166-2171). It is beyond the scope of our study to answer this question, but our most recent work (Karlstaedt A et al. *Circulation Research* 2020;126:60-74) together with Rossell S et al. indicates that regulation of pathway flux may originate from outside a given pathway.

Mechanistic insights: Our experimental design was aimed to elucidate whether there were common features of the metabolic alterations that could be causally related to Na elevation independent of whether this was induced acutely or chronically. We used an unbiased system-wide constraint-based modelling approach followed by hierarchical clustering and PCA analysis to better elucidate the pathways that are common. We were excited to discover that substrate preference can be ‘switched’ acutely and that it can be reversed by inhibiting NCLX with CGP. Our experimental and computational approaches are indeed just the first steps towards hypothesis building and achieving this goal.

The model is able to address these issues. We recently used CardioNet to describe the interplay between glycolytic intermediates and flux changes (see Karlstaedt A et al. *Circulation Research* 2020;126:60-74). The model was both in agreement with our experimental data and expanded our hypothesis. In the present study our modeling data are in agreement with the experimental results, clearly showing the common ‘fingerprint’ of acute and chronic Na elevation, and demonstrating the common systems-wide changes in substrate processing. Hopefully, future studies and modelling will be able to define the specific, as yet unknown, enzymes and pathways involved. But we would argue that the experimental design that should be performed to test the Ca dependence of these enzymes was not what we set out to achieve here and therefore is outside the scope of the current manuscript.

REVIEWERS' COMMENTS:

Reviewer #3 (Remarks to the Author):

The authors have replied to my prior remaining concerns and made efforts to better justify retention of the CardioNet modeling. They have improved the manuscript.

A couple of minor corrections that might otherwise be misleading:

Line 140: The CGP-induced increase in Palmitate oxidation in PLM3SA mice is labeled not significant in Fig 1A (and likewise for the apparent decrease in Glucose). Change wording and/or provide P values.

Line 161: "As to the PLM3SA data tissue NAD,..." perhaps should read "As for the PLM3SA data, tissue NAD,..."

RESPONSES TO REVIEWERS' COMMENTS:

Reviewer #3 (Remarks to the Author):

Responses in red

The authors have replied to my prior remaining concerns and made efforts to better justify retention of the CardioNet modeling. They have improved the manuscript.

Thank you.

A couple of minor corrections that might otherwise be misleading:

Line 140: The CGP-induced increase in Palmitate oxidation in PLM3SA mice is labeled not significant in Fig 1A (and likewise for the apparent decrease in Glucose). Change wording and/or provide P values.

CGP treated PLM 3SA hearts (red bar) have palmitate and glucose oxidation comparable to PLM WT hearts (white bar). There is a bar comparing these two data points that has NS above it.

The text has been amended as requested to clarify comparison (red font).

Line 161: "As to the PLM3SA data tissue NAD,..." perhaps should read "As for the PLM3SA data, tissue NAD,..."

Comma added.